# Oxidative phosphorylation is a pivotal therapeutic target of fibrodysplasia ossificans progressiva

Liping Sun[1], Yonghui Jin[1] , Megumi Nishio[2], Makoto Watanabe[3], Takeshi Kamakura[1], Sanae Nagata[2], Masayuki Fukuda[1], Hirotsugu Maekawa[2], Shunsuke Kawai[2], Takuya Yamamoto[4,5,6] , Junya Toguchida[1,2]

Heterotopic ossification (HO) is a non-physiological bone formation where soft tissue progenitor cells differentiate into chondrogenic cells. In fibrodysplasia ossificans progressiva (FOP), a rare genetic disease characterized by progressive and systemic HO, the Activin A/mutated ACVR1/mTORC1 cascade induces HO in progenitors in muscle tissues. The relevant biological processes aberrantly regulated by activated mTORC1 remain unclear, however. RNA-sequencing analyses revealed the enrichment of genes involved in oxidative phosphorylation (OXPHOS) during Activin A–induced chondrogenesis of mesenchymal stem cells derived from FOP patient–specific induced pluripotent stem cells. Functional analyses showed a metabolic transition from glycolysis to OXPHOS during chondrogenesis, along with increased mitochondrial biogenesis. mTORC1 inhibition by rapamycin suppressed OXPHOS, whereas OXPHOS inhibitor IACS-010759 inhibited cartilage matrix formation in vitro, indicating that OXPHOS is principally involved in mTORC1-induced chondrogenesis. Furthermore, IACS-010759 inhibited the muscle injury–induced enrichment of fibro/adipogenic progenitor genes and HO in transgenic mice carrying the mutated human ACVR1. These data indicated that OXPHOS is a critical downstream mediator of mTORC1 signaling in chondrogenesis and therefore is a potential FOP therapeutic target.

## Introduction

Fibrodysplasia ossificans progressiva (FOP, #135100; OMIM) is a rare genetic disease characterized by progressive and systemic heterotopic ossification in soft tissues such as skeletal muscles, tendons, ligaments, and fasciae (Shore & Kaplan, 2010). The clinical course of FOP is episodic and variable, characterized by alternating acute progressive and quiescent phases (Pignolo et al, 2016, 2019). Disease exacerbation often occurs after the emergence of painful soft tissue swellings, typically called flare-ups, which occur spontaneously or are triggered by trauma, infection, and medical or surgical procedures (Pignolo et al, 2016, 2019). Hence, heterotopic ossification (HO) in FOP patients is permanent and cumulative, resulting in progressive immobility with catastrophic complications (de Ruiter et al, 2021; Pignolo et al, 2022).

Histological examination of archival resected tissues of HO in FOP patients demonstrated that bone tissues are formed via the endochondral ossification process in which fibroblastic precursors residing in soft tissues undergo a sequential differentiation process from proliferating to hypertrophic chondrocytes, and finally, cartilage tissues are replaced by bone tissues (Kaplan et al, 1993; Shore & Kaplan, 2010). Most FOP patients harbor the c.617G>A (R206H) mutation in the ACVR1/ALK2 gene (FOP-ACVR1) (Shore et al, 2006; Kaplan et al, 2009; Zhang et al, 2013), which encodes one of the type I transmembrane receptors of bone morphogenetic proteins (BMPs), suggesting the involvement of BMP signaling in HO development. Using induced pluripotent stem cells (iPSCs) established from FOP patients (FOP-iPSCs), we identified Activin A as a crucial molecule that transduces the BMP signal from mutated ACVR1–containing receptor complexes in mesenchymal stem cells induced from FOP-iPSCs (FOP-iMSCs) (Hino et al, 2015). Identification of Activin A as a critical factor in HO was also reported using a conditional knock-in FOP mouse model (Hatsell et al, 2015). We also revealed that in combination with the TGF-β signal via its physiological receptor, ALK4, Activin A activates mechanistic target of rapamycin complex 1 (mTORC1) and drives FOP-iMSCs to chondrogenic differentiation in vitro, triggering HO in transgenic mice carrying a drug-inducible human FOP-ACVR1 gene (FOP-ACVR1 mice) (Hino et al, 2017). Because Activin A is a cytokine produced by inflammatory cells (Phillips et al, 2009), which infiltrate into injured tissues, it is a reasonable candidate for the flare-up factor that initiates the process of HO in FOP patients.

---

[1]Department of Regeneration Sciences and Engineering, Institute for Life and Medical Sciences, Kyoto University, Kyoto, Japan   [2]Department of Fundamental Cell Technology, Center for iPS Cell Research and Application, Kyoto University, Kyoto, Japan   [3]Life Science Research Center, Technology Research Laboratory, Shimadzu Corporation, Kyoto, Japan   [4]Department of Life Science Frontiers, Center for iPS Cell Research and Application, Kyoto University, Kyoto, Japan   [5]Institute for the Advanced Study of Human Biology (WPI-ASHBi), Kyoto University, Kyoto, Japan   [6]Medical-risk Avoidance Based on iPS Cells Team, RIKEN Center for Advanced Intelligence Project, Kyoto, Japan

Correspondence: jin.yonghui.4v@kyoto-u.ac.jp; togjun@cira.kyoto-u.ac.jp

A greater understanding of these pathological processes has facilitated many pharmacological studies, some of which are in clinical trials (Meng et al, 2022), targeting each step of HO, including a neutralizing antibody against Activin A (Hatsell et al, 2015; Di Rocco et al, 2023), kinase inhibitors for ACVR1 (Cuny et al, 2008; Yu et al, 2008; Mohedas et al, 2013; Sanvitale et al, 2013; Williams et al, 2021), *ACVR1* mRNA down-regulation (Kaplan et al, 2012; Takahashi et al, 2012; Cappato et al, 2016; Maruyama et al, 2022), mTORC1 inhibition (Hino et al, 2017), modifying differentiation by RARγ activation (Shimono et al, 2011; Pignolo et al, 2023), and, most recently, AAV-based gene therapy targeting mutant ACVR1 using miRNA (Yang et al, 2022). In addition, therapies targeting the inflammation or microenvironment of local lesions, such as depletion of macrophages (Convente et al, 2018) and inhibition of c-KIT on mast cells using a multi-kinase inhibitor (tested in a clinical trial [Kaplan et al, 2021]), have been examined. Using monocytes induced from FOP-iPSCs, we previously identified a possible therapeutic target, LYVE1, an Activin A–regulated factor involved in matrix formation (Maekawa et al, 2022).

Among the different therapeutic targets, we have been mainly focusing on mTORC1 signaling. Using FOP-ACVR1 mice, we established a model of injury-induced HO in which the pinching of leg muscles induced HO at both injured and uninjured sites, suggesting locally produced Activin A may have systemic effects (Maekawa et al, 2020). Prophylactic administration of rapamycin effectively inhibited HO, thereby confirming the role of mTORC1 signaling in HO (Maekawa et al, 2020). It is still unclear, however, how activated mTORC1 induces precursors for chondrogenic differentiation. To improve mTORC1 inhibition as a potential therapy, a better understanding of the biological processes downstream of mTORC1 leading to HO is critical.

mTORC1 plays a central role by connecting multiple environmental signals, such as nutrient and growth factor availability and stress, to metabolic processes to optimize cellular metabolism (Zoncu et al, 2011). Rapamycin-sensitive mTORC1 substrates include S6K1 (ribosomal protein S6 kinase 1) and 4E-BP1 (eukaryotic initiation factor 4E-binding protein 1), both of which regulate cap-dependent translation, thus allowing mTORC1 to increase protein synthesis and ribosomal biogenesis (Zoncu et al, 2011). These biochemical activations affect multiple metabolic processes, including cell growth, lipogenesis, autophagy, and angiogenesis (de la Cruz López et al, 2019). Mitochondrial biogenesis is also activated by mTORC1 signaling (Zoncu et al, 2011), up-regulating energy production through enhanced oxidative phosphorylation (OXPHOS). Here, we demonstrate that OXPHOS is functionally involved in Activin A–induced FOP-iMSC chondrogenic differentiation in vitro and injury-induced HO in association with increased mitochondrial biogenesis of platelet-derived growth factor receptor alpha (PDGFRα)–positive cells, a marker of fibro/adipogenic precursors (FAPs). Moreover, IACS-010759 (hereafter called IACS), a clinically applicable OXPHOS blocker targeting mitochondria complex I (Molina et al, 2018; Yap et al, 2023), inhibited the chondrogenesis of FOP-iMSCs in vitro and suppressed HO in vivo, indicating that OXPHOS is a novel metabolic therapeutic target for FOP treatment.

# Results

## Transcriptomic and metabolomic profiles implicate the involvement of OXPHOS in Activin A–induced chondrogenesis of FOP-iMSCs

To evaluate the chondrogenic differentiation of iMSCs in two-dimensional (2D) cultures, we employed the monolayer induction method with some modifications (Ruhl & Beier, 2019). Activin A treatment gradually increased the expression of genes related to chondrogenic differentiation (*SOX9*, *COL2A1*, and *ACAN*) in FOP-iMSCs, but not in the mutation-rescued iMSCs (resFOP-iMSCs) (Fig 1A). At day 9, FOP-iMSCs produced an Alcian Blue–positive matrix, which was not observed in resFOP-iMSCs (Fig 1B). These results indicated that the current 2D method successfully reproduced previous results using the 2D micromass method (Hino et al, 2015).

With this experimental system, we performed transcriptome analysis before (day 0) and after (day 6) chondrogenesis by bulk RNA sequencing, followed by gene set enrichment analysis (GSEA). Consistent with our previous findings (Hino et al, 2017), mTORC1 signaling was identified in the list of enriched gene sets by comparing the data of FOP-iMSCs at day 6 with those at day 0 (Fig 1C) or with those of resFOP-iMSCs at day 6 (Fig 1D). The OXPHOS gene set was also identified to be enriched by similar comparisons (Fig 1C and D). In contrast, neither mTORC1 signaling nor OXPHOS was identified as enriched gene sets in resFOP-iMSCs after chondrogenic induction with Activin A (Fig 1E). Consistent with the GSEA results, the expression of complex I subunits in the electron transport chain was up-regulated after chondrogenic induction in FOP-iMSCs, but not in resFOP-iMSCs (Fig 1F). Similar up-regulation was observed in subunits of complexes III, IV, and V (Fig S1).

The relative amount of TCA cycle metabolites gradually increased during chondrogenesis of FOP-iMSCs, and some of them, such as glucose 6-phosphate, pyruvate, 2-ketoglutarate, fumarate, and malate, had increased significantly by day 9 (Fig 1G). These changes were, however, not observed in resFOP-iMSCs (Fig 1G). Interestingly, the relative amount of glucose, the preferential fuel of the TCA cycle, was also increased during the chondrogenesis of FOP-iMSCs (Fig 1G). These results indicated that OXPHOS was activated in association with mTORC1 signaling during chondrogenesis of FOP-iMSCs.

## OXPHOS activity and mitochondrial biogenesis were enhanced during Activin A–induced chondrogenesis of FOP-iMSCs

To confirm the functional activation of OXPHOS, oxygen consumption rate (OCR) and extracellular acidification rate (ECAR) representing mitochondrial respiration and glycolysis, respectively, were measured.

OCR was quantified using the Seahorse XF analyzer, in conjunction with XF Mito Stress Kit, which evaluates the overall mitochondrial respiration capacity. By sequentially adding oligomycin, FCCP, and rotenone/antimycin A, we were able to measure basal respiration, ATP production, and maximal respiration. Basal

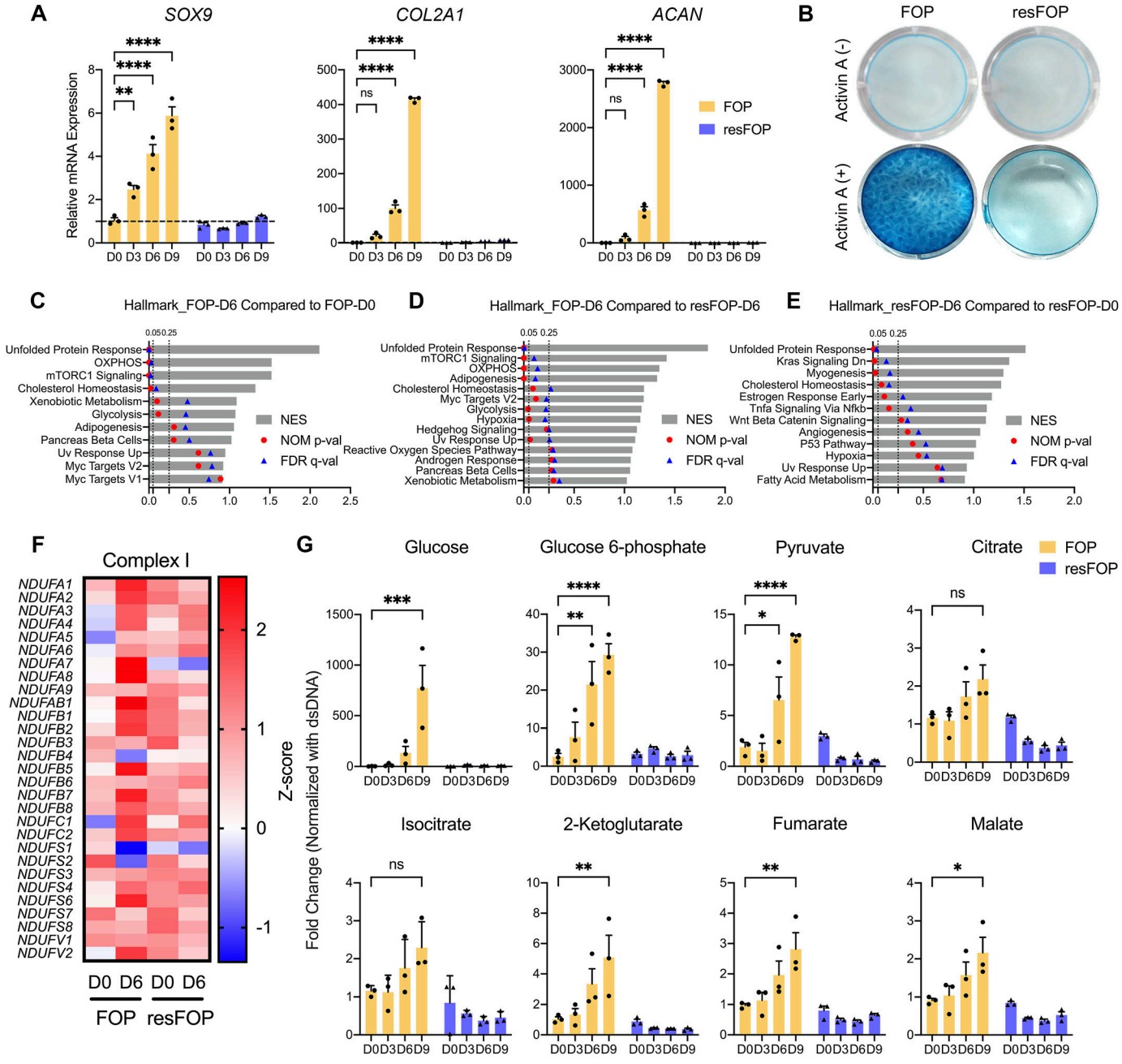

**Figure 1. Transcriptomic and metabolomic profiles implicate the involvement of OXPHOS during Activin A–induced chondrogenesis of FOP-iMSCs.**
**(A)** mRNA expression of chondrogenic markers (*SOX9*, *COL2A1*, and *ACAN*) during chondrogenesis. RNAs were extracted from cells at each time point and assessed by RT–qPCR. Expression levels were normalized to FOP-iMSCs at day 0. Each datum is presented as a dot with the mean value ± SEM. *n* = 3 biological replicates. **(B)** Alcian Blue staining at day 9 of chondrogenesis. **(C, D, E)** List of enriched gene sets. **(C, D, E)** RNA-sequencing data were compared between those of FOP-iMSCs at day 6 and day 0 (C), FOP-iMSCs at day 6 and resFOP-iMSCs at day 6 (D), and resFOP-iMSCs at day 6 and day 0 (E). NES, normalized enrichment score; NOM p-val, nominal *P*-value, < 0.05 is significant; FDR q-val, false discovery rate q-value, < 0.25 is significant. **(F)** Heatmap of complex I subunit genes. Data are presented by colors based on the Z-score of each gene. **(G)** OXPHOS-related intracellular metabolite analysis by gas chromatography–mass spectrometry. *n* = 3 biological replicates. *P < 0.05, **P < 0.01, ***P < 0.001, and ****P < 0.0001, as determined by two-way ANOVA (Tukey's multiple comparisons test).

respiration reflects the cell's energy demand under baseline conditions. ATP production indicates the amount of ATP generated by the mitochondria, contributing to the cell's energy requirements. Maximal respiration represents the cell's peak respiratory rate achievable under the given conditions.

In FOP-iMSCs, basal mitochondrial respiration, maximal mitochondrial respiration, and ATP production were significantly increased after chondrogenic induction with Activin A, whereas those parameters did not change in resFOP-iMSCs despite Activin A treatment (Fig 2A). In contrast, ECAR declined in both FOP-iMSCs

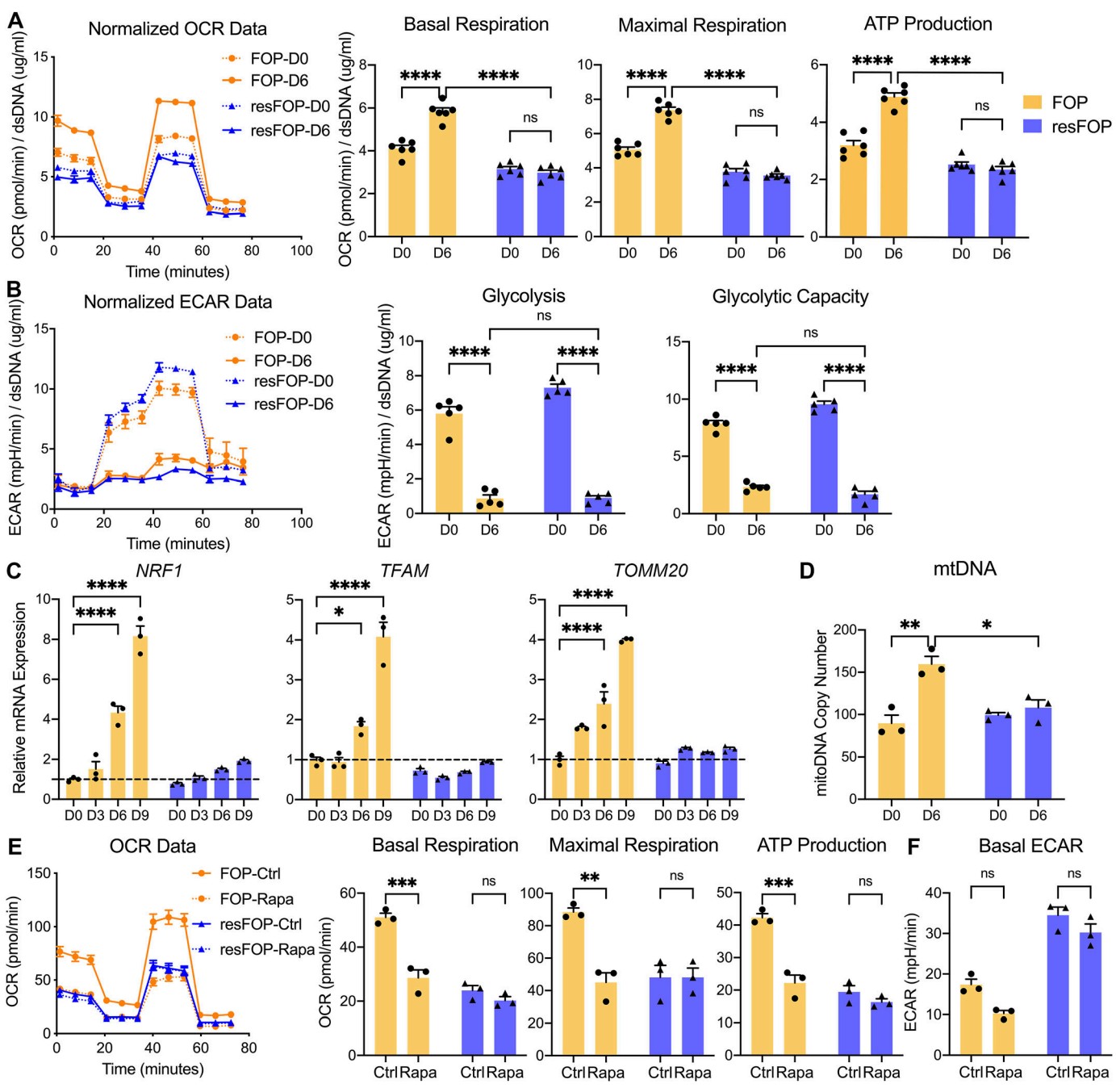

**Figure 2. mTORC1 signaling enhances OXPHOS activity and mitochondrial biogenesis during Activin A–induced chondrogenesis of FOP-iMSCs.**
**(A)** FOP- and resFOP-iMSCs were induced for chondrogenesis for 6 d and then processed for OCR analyses followed by consecutive injections of oligomycin ($2 \mu M$), FCCP ($1 \mu M$), and antimycin A ($0.5 \mu M$)/rotenone ($0.5 \mu M$). Basal and maximum respiration and ATP production were calculated and are presented as dots with the mean value ± SEM. $n$ = 6 biological replicates. **(B)** Glycolysis stress test. FOP- and resFOP-MSCs were induced for chondrogenesis for 6 d and then processed for the ECAR analyses followed by consecutive injections of glucose (10 mM), oligomycin ($2 \mu M$), and 2-DG (50 mM). Each datum is presented as a dot with the mean value ± SEM. $n$ = 5 biological replicates. **(C)** Expression of genes related to mitochondrial biogenesis. RNAs were extracted at each time point and analyzed by RT–qPCR. The expression level was normalized to FOP-iMSCs at day 0. Each datum is presented as a dot with the mean value ± SEM. $n$ = 3 biological replicates. **(D)** Quantitative analysis of the mitochondrial DNA copy number. DNAs were extracted at each time point and analyzed with Human Mitochondrial DNA Monitoring Primer Set. Each datum is presented as a dot with the mean value ± SEM. $n$ = 3 biological replicates. **(E, F)** Effect of rapamycin on OXPHOS. **(A, B, E, F)** FOP- and resFOP-iMSCs were induced for chondrogenesis for 4 d, then cultured with or without rapamycin (40 nM) for 24 h, and then processed for OCR analyses (E) as described in (A) and the ECAR analyses (F) as described in (B). Each datum is presented as a dot with the mean value ± SEM. $n$ = 3 biological replicates. *$P$ < 0.05, **$P$ < 0.01, ***$P$ < 0.001, and ****$P$ < 0.0001, as determined by two-way ANOVA (Tukey's multiple comparisons test).

and resFOP-iMSCs after chondrogenic induction (Fig 2B). These data suggest energy metabolism shifted from glycolysis to OXPHOS during chondrogenesis of FOP-iMSCs. Mitochondrial biogenesis is a process necessary to increase mitochondrial activity. The expression of mitochondria transcription factor A (*TFAM*) and nuclear respiratory factor 1, essential for transcription and replication of mitochondrial DNA (mtDNA) (de la Cruz López et al, 2019), and translocase of outer mitochondrial membrane 20 (*TOMM20*), a member of the preprotein translocase complex in the mitochondrial membrane (Bohnert et al, 2007), was elevated significantly during chondrogenesis of FOP-iMSCs, but not resFOP-iMSCs (Fig 2C). In addition, mtDNA content increased in FOP-iMSCs after chondrogenesis (Fig 2D). Altogether, these data indicate Activin A–driven chondrogenesis stimulates mitochondrial biogenesis and activates the OXPHOS pathway.

Because GSEA showed coactivation of mTOR signaling and OXPHOS during Activin A–induced chondrogenesis of FOP-iMSCs, we explored whether mTORC1 signaling plays a causative role in the activation of OXPHOS in FOP-iMSCs by inhibiting it using rapamycin. Rapamycin treatment for 24 h significantly decreased each OCR parameter in FOP-iMSCs (Fig 2E), whereas no effect was observed in ECAR (Fig 2F). Protein synthesis serves as a downstream effector of mTORC1 signaling. The suppression of protein synthesis using cycloheximide or puromycin for a duration of 24 h led to a notable reduction in OCR parameters (Fig S2A and B). This suggests that protein synthesis is one of the biological processes connecting mTORC1 signaling to OXPHOS. Although the detailed molecular mechanism is unclear, these results indicate that the Activin A/FOP-ACVR1/mTORC1 cascade is responsible for FOP-specific OXPHOS activation.

### Inhibition of OXPHOS suppresses chondrogenesis of FOP-iMSCs

To confirm the functional role of OXPHOS activation in aberrant chondrogenesis of FOP-iMSCs, OXPHOS was inhibited during chondrogenesis by IACS, which blocks cellular respiration through the inhibition of complex I (Molina et al, 2018). The effects of IACS on OXPHOS in FOP-iMSCs were examined by OCR and ECAR analyses (Fig S3A and B). Its effects on chondrogenic differentiation were evaluated using both 2D and 3D inductions. In 2D induction, IACS treatment for 9 d resulted in a dose-dependent inhibition of nodule formation (Fig 3A) and Alcian Blue staining (Fig 3B) in FOP-iMSCs. Although slight growth inhibition was observed in FOP-iMSCs at concentrations of 10 nM or higher (Fig 3C), the inhibition of GAG production per DNA indicated the effects of IACS on chondrogenic differentiation were not due to cytocidal effects (Fig 3D). Histological examination of 3D pellets showed marked reductions in Safranin O staining and pellet size (Fig 3E), which were caused by inhibitory effects on matrix production, not growth (Fig 3F and G). This suppression of chondrogenesis by IACS was reproducibly observed in other FOP-iMSC clones (Fig S3C). This critical role of OXPHOS in chondrogenesis of FOP-iMSCs was confirmed using another mitochondrial complex I inhibitor metformin (Fig S4). The parameters of OCR were dose-dependently decreased by metformin treatment for 24 h (Fig S4A). Furthermore, metformin dramatically inhibited the chondrogenesis of FOP-iMSCs (Fig S4B). Both IACS and metformin did not affect the mTORC1 signaling (Fig S4C),

indicating that inhibiting complex I does not influence mTORC1 activity. These data indicate OXPHOS activation is causatively involved in Activin A–induced chondrogenesis of FOP-iMSCs.

### Inhibition of OXPHOS suppresses heterotopic cartilage formation in FOP-ACVR1 mice

To investigate whether OXPHOS activation was involved in HO, the effects of IACS were analyzed using an injury-induced HO model, in which HO was induced by pinching the gastrocnemius muscles (PINCH) of transgenic mice carrying a doxycycline (DOX)-inducible human ACVR1^R206H gene (Maekawa et al, 2020). Mice were randomly assigned into four groups: DOX (−)/PINCH (−), DOX (−)/PINCH (+), DOX (+)/PINCH (+), and DOX (+)/PINCH (+)/IACS (+) (Fig 4A). DOX was administered to mice in DOX (+) groups from 7 d before PINCH, and mice in the IACS (+) group were treated with IACS from 2 d before PINCH.

We first evaluated the effects of IACS on chondrogenesis. Based on our previous experiments (Maekawa et al, 2020), 7 d after injury was the most appropriate time point for evaluating cartilage formation without obvious bone formation. Entire gastrocnemius muscles were harvested on day 7 and processed for RNA-sequencing analyses or histological examination. RNA-sequencing analyses identified that the chondrocyte differentiation gene set was enriched in DOX (+)/PINCH (+) samples when compared to DOX (−)/PINCH (+) samples (Fig 4B). Heatmap analyses of each sample showed that PINCH up-regulated genes related to chondrocyte differentiation in DOX (+)/PINCH (+), which were down-regulated by IACS treatment (Fig 4C). By histological examination, PINCH induced centrally nucleated myofibers expressing myozenin 1, which is one of the regeneration-related proteins (Yoshimoto et al, 2020) (Fig S5) in the DOX (−)/PINCH (+) sample (Fig 4D, a–f). In the DOX (+)/PINCH (+) sample, cartilage tissues stained by Safranin O were found and were surrounded by fibroblastic cells (Fig 4D, g–i). Fibroblastic cells accumulated in injured tissues of IACS (+) samples, in which no cartilage tissues were found (Fig 4D, j–l), suggesting that IACS inhibited chondrocyte differentiation from precursors.

Consistent with the in vitro data, chondrocyte differentiation was associated with the activation of mTORC1 signaling (Fig S6A and C), whereas the effects of IACS treatment on mTORC1 signaling were minimal (Fig S6B and C). This result indicated that the inhibitory effects of IACS on chondrocyte differentiation were not through mTORC1 inhibition but via downstream events. This is consistent with the results in vitro (Fig S4C). The gene sets of abnormal activity of mitochondria and decreased activity of mitochondrial complex I or IV were enriched by comparing the data of DOX (+)/PINCH (+)/IACS (+) with those of DOX (+)/PINCH (+) (Fig S6D), indicating that IACS treatment inhibited mitochondrial activity in vivo, resulting in the inhibition of chondrocyte differentiation.

### Inhibition of OXPHOS suppresses heterotopic bone formation in FOP-ACVR1 mice

To confirm that the inhibition of cartilage formation by IACS results in the inhibition of HO, DOX (+)/PINCH (+) and DOX (+)/PINCH (+)/IACS (+) mice were followed until day 21, and HO was evaluated via

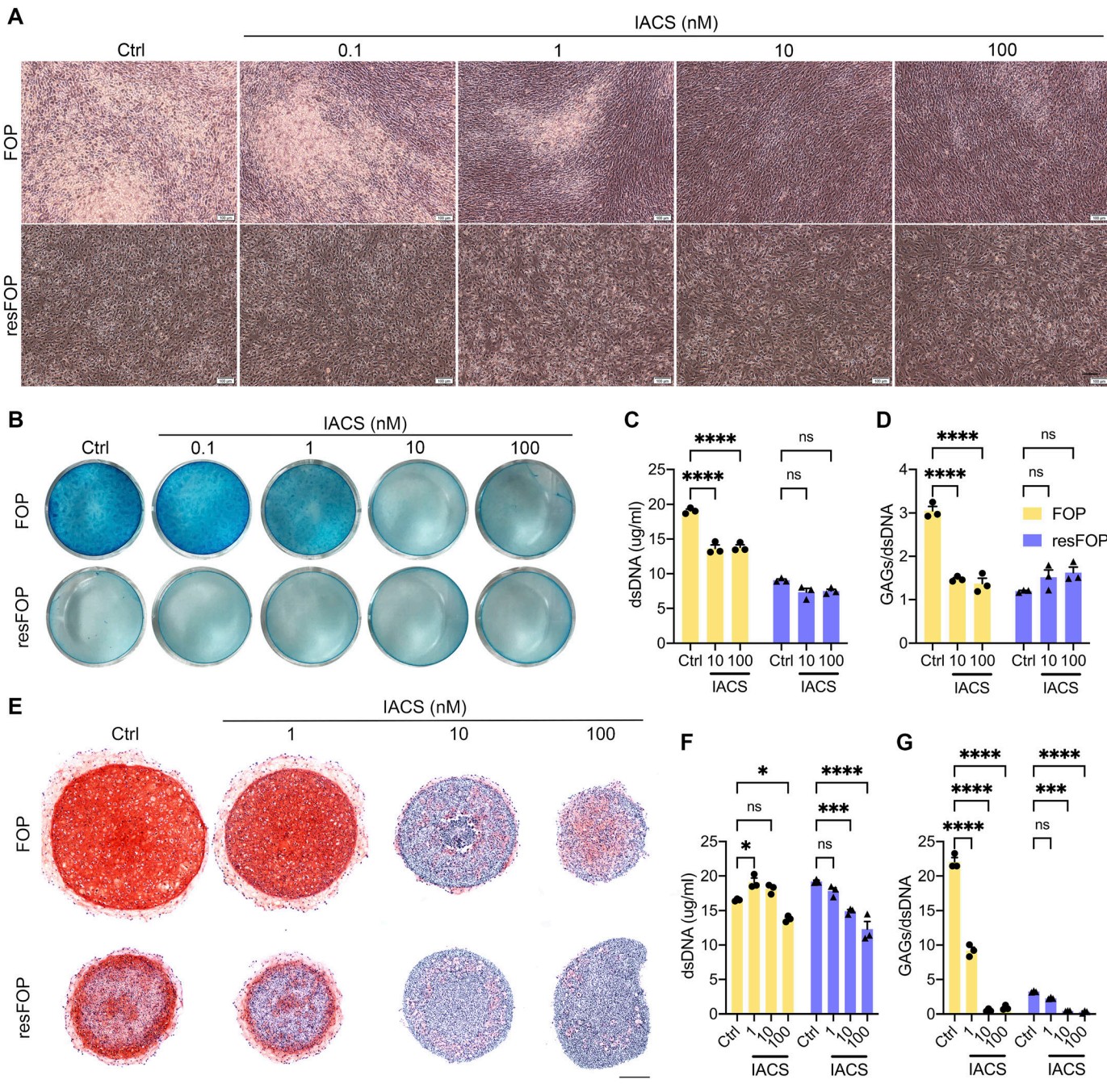

**Figure 3. Inhibition of OXPHOS suppresses chondrogenesis of FOP-iMSCs.**
**(A, B, C, D)** 2D induction of chondrogenesis with IACS. **(A, B)** Phase-contrast images (A) and Alcian Blue staining photos (B) of FOP- or resFOP-iMSCs cultured with indicated concentrations of IACS for 9 d. dsDNA and GAG were extracted at the end of induction, and their concentrations were measured. **(C, D)** Data are presented as the concentration of dsDNA (C) and the ratio of GAG to dsDNA (D). Each datum is presented as a dot with the mean value ± SEM. $n$ = 3 biological replicates. **(E, F, G)** 3D induction of chondrogenesis with IACS. **(E)** Chondrocyte pellets were sliced and stained by Safranin O and Fast Green (E). dsDNA and GAG were extracted at the end of induction, and the concentrations of them were measured. **(F, G)** Concentration of dsDNA (F) and the ratio of GAG to dsDNA (G). Each datum is presented as a dot with the mean value ± SEM. $n$ = 3 biological replicates. *$P$ < 0.05, **$P$ < 0.01, ***$P$ < 0.001, and ****$P$ < 0.0001, as determined by two-way ANOVA (Tukey's multiple comparisons test). Scale bar = 100 $\mu$m in (A), and scale bar = 200 $\mu$m in (E).

micro-computed tomography ($\mu$CT) (Fig 5A). All 12 mice in the DOX (+)/PINCH (+) group exhibited HO, whereas IACS treatment suppressed HO significantly, and the suppression effect was complete in some mice (Figs 5B and C and S7A and B). As in our previous study, we again observed that PINCH sometimes induced HO in contralateral limbs (Fig S7A and B), and IACS treatment reduced the incidence of this event completely (Fig 5D). Notably, the expression of *Inhba*, a gene encoding one of the Activin A subunits, was

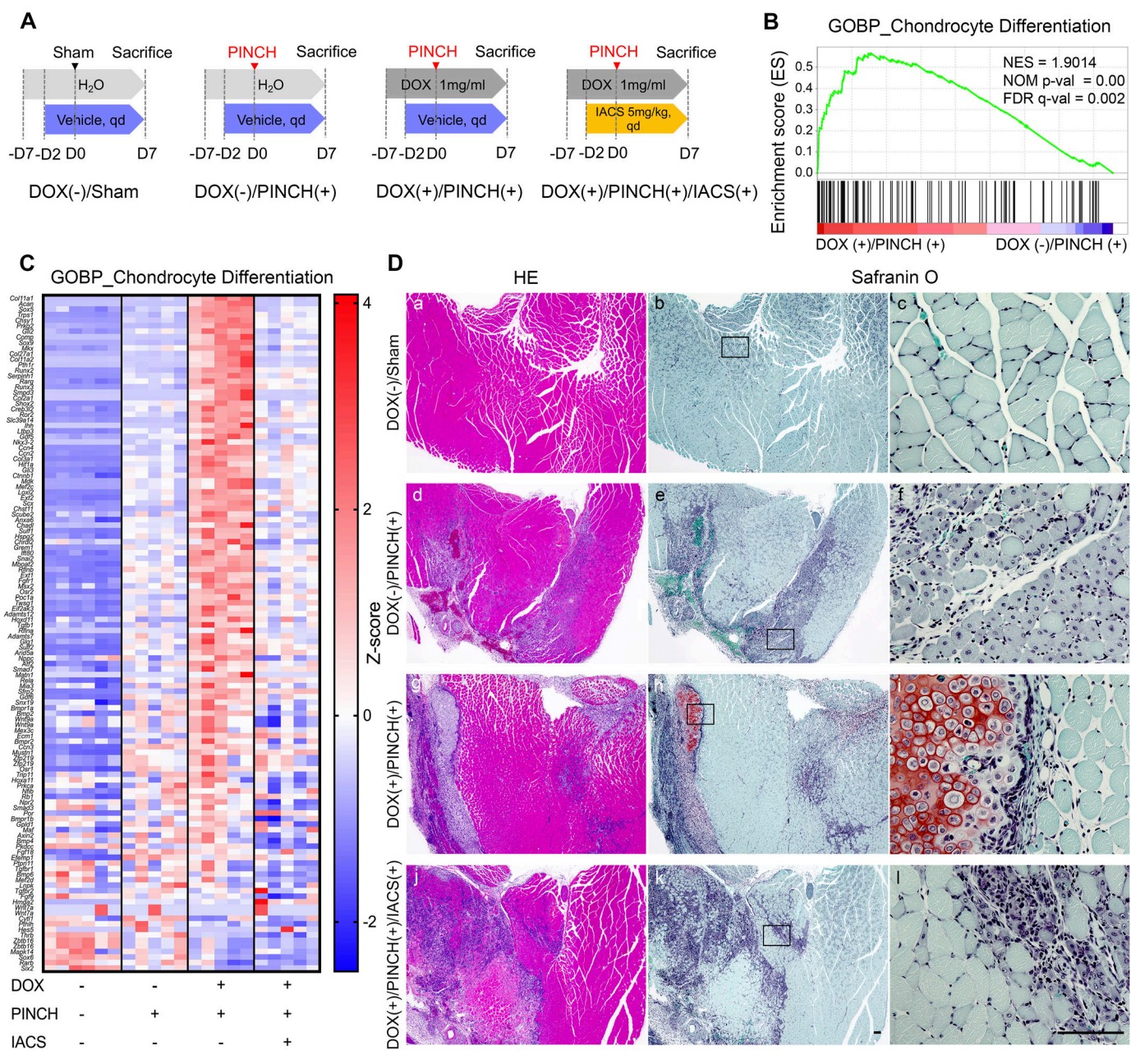

**Figure 4. Inhibition of OXPHOS suppresses heterotopic cartilage formation in FOP-ACVR1 mice.**
**(A)** Schematic diagram of experiments. **(B)** Enrichment plot for the gene set of GOBP_Chondrocyte Differentiation. The plot was drawn by the comparison between the data of DOX (+)/PINCH (+) and DOX (−)/PINCH (+). **(C)** Heatmap of genes listed in the gene set of GOBP_Chondrocyte Differentiation. Data are presented by colors based on the Z-score of each gene. **(D)** Histological examination of resected muscle tissues. Tissues were stained by HE (a, d, g, j) or Safranin O and Fast Green (b, e, h, k). The boxed areas are shown as magnified views (c, f, i, l). Scale bar = 100 μm.

significantly up-regulated in injury sites of DOX (+)/PINCH (+) mice at day 7 and was dramatically inhibited by IACS administration (Fig 5E). This reduction of *Inhba* expression might be responsible for diminishing HO formation in contralateral limbs. IACS treatment showed no apparent effects on body weight compared with the vehicle (Fig 5F).

Histological analysis of injury sites in the DOX (+)/PINCH (+) group showed mature bone tissues (Fig 5G, a and b) consisting of Safranin O–positive cartilage tissues (Fig 5G, c and d) and von

Kossa–positive calcified tissues (Fig 5G, e and f), indicative of endochondral ossification. Safranin O–positive (Fig 5H, c and d) and von Kossa–positive calcified areas were rarely found (Fig 5H, e and f) in the DOX (+)/PINCH (+)/IACS (+) group (Fig 5H, a and b). Treatment with IACS showed no obvious effects on vital organs during these experiments (Fig S8A and B). These data indicate that the inhibition of cartilage formation by IACS also suppressed bone formation and represents a potential therapeutic approach for FOP.

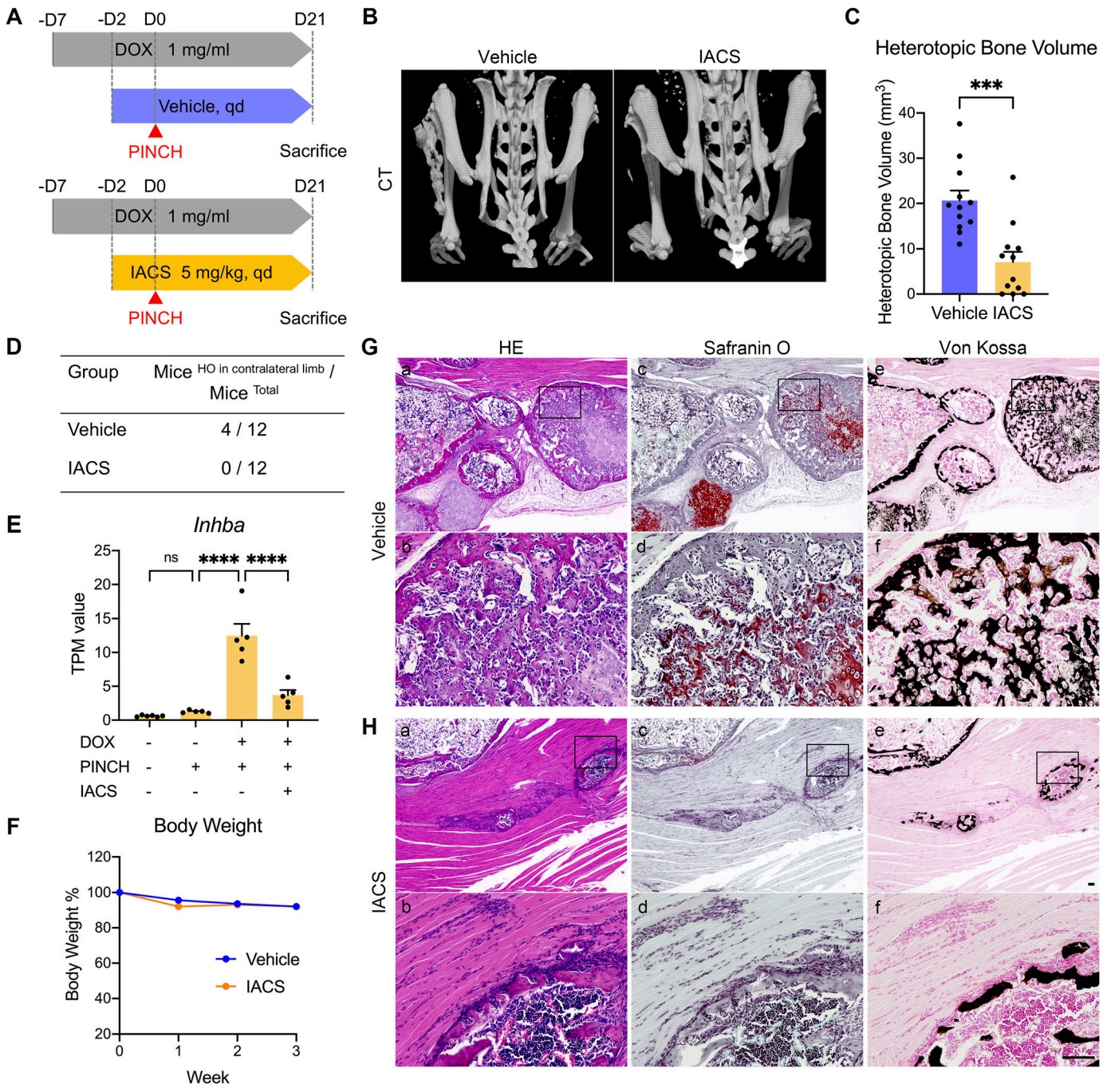

**Figure 5. Inhibition of OXPHOS suppresses heterotopic bone formation in FOP-ACVR1 mice.**
**(A)** Schematic view of experiments. **(B)** Representative images of μCT 21 d after pinch. **(C)** Quantification of HO volumes at pinch sites. HO volumes were measured by μCT. All data are presented as dots with the mean ± SEM. *n* = 12 mice. **(D)** Number of mice with HO in contralateral limbs. **(E)** mRNA expression of *Inhba* gene from RNA sequencing. Expression levels are shown as TPM (Transcripts Per Kilobase Million). **(F)** Body weight change during 21 d with or without IACS administration. **(G, H)** Histological examination of resected tissues. Tissues were stained by HE (a, b), or Safranin O and Fast Green (c, d), or von Kossa (e, f). Boxed areas in (a, c, e) are shown as magnified views (b, d, f). Scale bar = 100 μm. \*\*\*P < 0.001 and \*\*\*\*P < 0.0001, as determined by two-way ANOVA (Tukey's multiple comparisons test).

## Inhibition of OXPHOS suppresses the differentiation of PDGFRα-positive cells

IACS inhibited the chondrogenic differentiation of FOP-iMSCs in vitro and the ectopic cartilage formation of FOP-ACVR1 mice in vivo, suggesting IACS may inhibit chondrogenic differentiation of precursors in injured muscle tissues. FAPs, muscle-resident progenitors, a primary cell-of-origin of HO in the knock-in mouse model of FOP (Lees-Shepard et al, 2018), are characterized by the expression of PDGFRα (Uezumi et al, 2010). FOP-iMSCs expressed

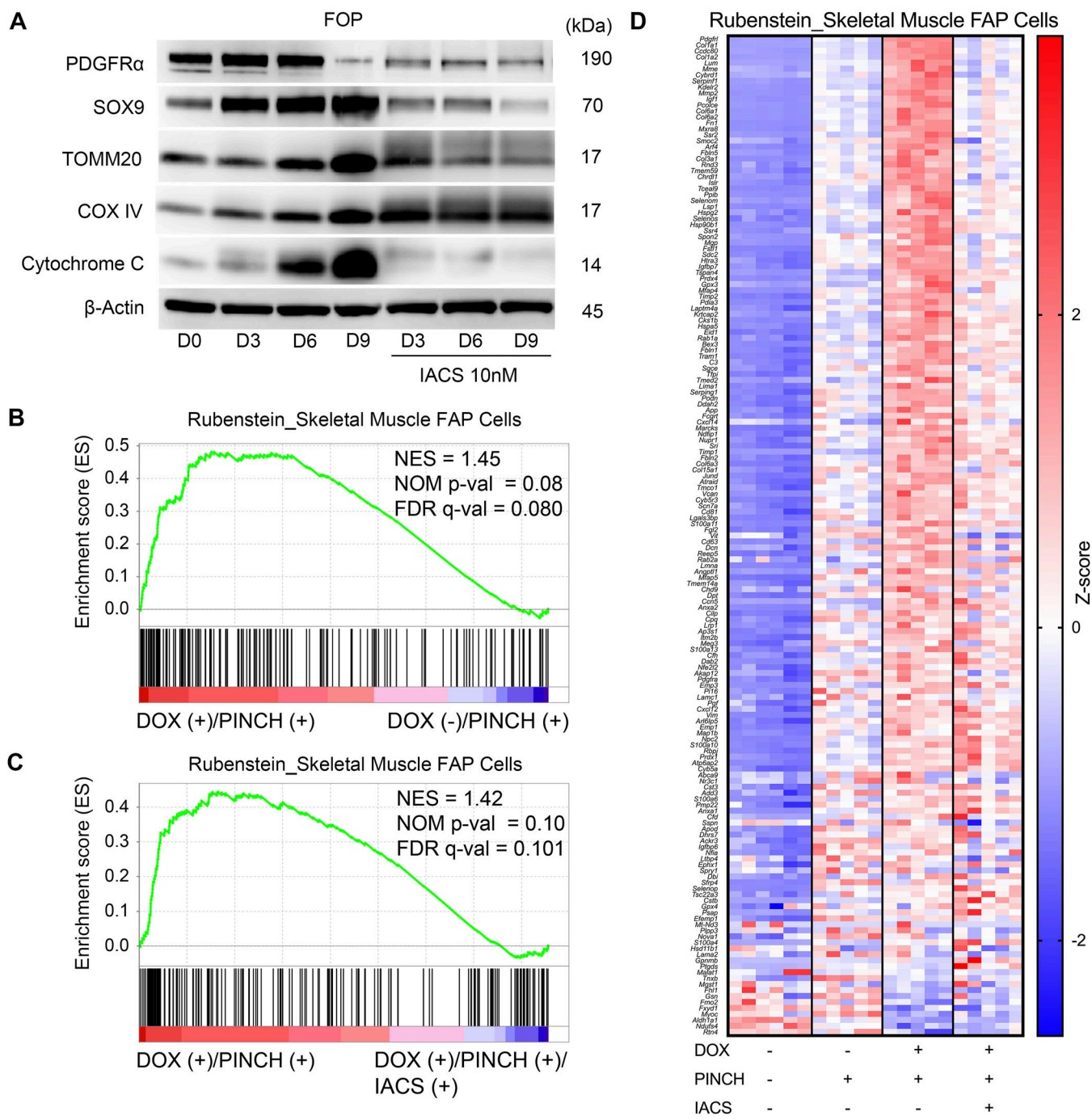

**Figure 6. Inhibition of OXPHOS suppresses the proliferation and differentiation of PDGFRα-positive cells: molecular analyses.**
**(A)** Immunoblotting analysis of PGDFRα, SOX9, TOMM20, COX IV, and cytochrome C. Proteins were extracted from FOP-iMSCs at each time point during chondrogenesis with or without IACS treatment (10 nM). **(B, C)** Enrichment plot of the gene set of Rubenstein_Skeletal Muscle FAP Cells. **(B, C)** Plot was created by the comparison between the data of DOX (+)/PINCH (+) and DOX (−)/PINCH (+) (B), and DOX (+)/PINCH (+) and DOX (+)/PINCH (+)/IACS (+) (C). **(D)** Heatmap of genes listed in the gene set of Rubenstein_Skeletal Muscle FAP Cells. Data are presented by colors based on the Z-score of each gene.

PDGFRα under standard culture conditions, which decreased after chondrogenic differentiation (Fig 6A). In contrast, expression levels of SOX9, TOMM20, COX IV, and cytochrome C gradually increased during induction (Fig 6A), suggesting that PDGFRα-positive FOP-iMSCs differentiate to SOX9-positive chondrogenic cells in association with the up-regulation of OXPHOS. IACS treatment

decreased PDGFRα expression by FOP-iMSCs and inhibited the up-regulation of SOX9- and OXPHOS-related genes (Fig 6A). The involvement of FAPs in injury-induced HO in FOP-ACVR1 mice was investigated by RNA-sequencing and immunohistochemical analyses. The gene set of skeletal muscle FAPs was enriched in the DOX (+)/PINCH (+) group by the comparison with the DOX (−)/PINCH (+)

group (Fig 6B), and IACS treatment inhibited the enrichment of this gene set (Fig 6C). The heatmap showed that genes related to FAPs were up-regulated by PINCH and further enhanced by DOX (+), which were down-regulated by IACS treatment (Fig 6D). As the gene set characterizing FAPs was derived from single-cell RNA sequencing of mononuclear cells from skeletal muscle (Rubenstein et al, 2020), the representation of this gene signature in bulk RNA sequencing reflects the abundance of FAPs. Thus, it can be inferred that PINCH potentially increased the FAP population, further augmented by DOX (+), and subsequently diminished by IACS administration. These data suggest the involvement of FAPs in ectopic cartilage formation in FOP-ACVR1 mice.

To further investigate the involvement of FAPs in HO, we searched for PDGFRα-positive cells in resected tissues. No cartilage or calcified tissues were observed in DOX (−)/PINCH (+) samples (Fig S9A), and PDGFRα-positive cells were rarely found (Fig S9B). In contrast, regenerated myofibers strongly expressed mitochondrial markers TOMM20, SDHA, and COX IV (Fig S9B), consistent with a previous report revealing OXPHOS to be crucial in skeletal muscle regeneration (Hong et al, 2022). In DOX (+)/PINCH (+) samples, PDGFRα-positive cells accumulated adjacent to the Safranin O–positive area (Fig 7A, a–c and c") where SOX9-positive cells were found (Fig 7A, d and d"). Both cell types highly expressed TOMM20 (Fig 7A, e and e"). In the area showing cartilage tissues (Fig 7B), PDGFRα-positive cells were found in fibroblastic cell clusters surrounding cartilage tissues (Fig 7B, a–c and c"), in which cells were SOX9-positive (Fig 7B, d and d"). Both PDGFRα- and SOX9-positive cells were strongly positive for TOMM20 (Fig 7B, e and e"). These data suggested that mitochondrial biogenesis was markedly enhanced during chondrogenesis from progenitors to differentiated cells. The treatment with IACS did not affect the accumulation of fibroblastic cells (Fig 7C, a and b) but dramatically inhibited the appearance of TOMM20-positive cells (Fig 7C, e and e"), which were associated with the decrease in PDGFRα-positive cells (Fig 7C, c and c") and cartilage formation by SOX9-positive cells (Fig 7C, d and d").

The impact of IACS on apoptosis in PDGFRα-positive cells (Fig S10A, a and a" and B, a and a") was assessed through cleaved caspase-3 staining. Few cells exhibited positivity for cleaved caspase-3 in DOX(+)/PINCH(+) samples (Fig S10A, b and b"), and there was no notable increase in positively stained cells with IACS administration (DOX [+]/PINCH[+]/IACS[+]) (Fig S10B, b and b"). In addition, IACS administration showed no apparent differences in Ki-67–positive cells (Fig S10A, c and c" and B, c and c") or phospho-histone H3–positive cells (Fig S10A, d and d" and B, d and d"). In summary, IACS appears to inhibit the chondrogenic differentiation of PDGFRα-positive cells without significantly influencing cell death or proliferation.

In summary, the aberrant chondrogenesis driven by Activin A in FOP was initiated by the activating signal from mTORC1 and transduced to the mitochondria to enhance OXPHOS via a boost in mitochondrial biogenesis. Thus, OXPHOS suppression may be a suitable therapeutic approach for diminishing heterotopic cartilage formation in FOP.

## Discussion

Here, we demonstrated that OXPHOS is a critical metabolic pathway aberrantly activated by mTORC1 signaling to promote chondrogenic

differentiation of FOP-iMSCs in vitro and injury-induced heterotopic chondrogenesis and HO in vivo. GSEA identified the enrichment of several pathways in addition to mTORC1 signaling during chondrogenesis of FOP-iMSCs. The enrichment of genes involved in the unfolded protein response is unsurprising because mTORC1 signaling up-regulates protein synthesis by enhancing protein translation and ribosome biogenesis, which may induce endoplasmic reticulum stress (Reiling & Sabatini, 2008). The enrichment of genes associated with cholesterol homeostasis is of interest as it was recently shown to be a crucial biological process in the chondrogenic differentiation of MSCs (Tsushima et al, 2018). mTORC1 regulates SREBP1, a critical transcription factor for cholesterol homeostasis, directly by controlling its nuclear translocation and indirectly by the phosphorylation of S6K1 (Simcox & Lamming, 2022). These results suggest that mTORC1 signaling may initiate chondrogenic differentiation through several distinct mechanisms. In particular, we focused on the role of OXPHOS, which was shown to determine the cell fate of stem cells in several lineages (Beckervordersandforth, 2017; Papa et al, 2019; Mohammadalipour et al, 2020).

The functional relationship between mTORC1 and mitochondrial activity has been analyzed in several aspects, which demonstrated that mTORC1 up-regulated mitochondrial biogenesis (Simcox & Lamming, 2022). Mitochondrial biogenesis is the process through which cells increase mitochondrial numbers, resulting in the up-regulation of metabolic enzymes involved in glycolysis and OXPHOS, thereby increasing OXPHOS activity (Popov, 2020). Several regulatory mechanisms of mitochondrial biogenesis by mTORC1 have been reported, including 4E-BP–dependent translational regulation and PGC1α-dependent transcriptional regulation of mitochondria-related genes such as TFAM and NFR1 (de la Cruz López et al, 2019). In this study, we observed mitochondrial gene upregulation and increased mtDNA copy number during Activin A–induced chondrogenesis in vitro and increased TOMM20 expression in vivo, together suggesting the possibility of the involvement of the mTORC1/mitochondrial biogenesis/OXPHOS cascade during chondrogenesis and HO in FOP. Because the resection of HO tissues of FOP patients is contraindicated, it is difficult to confirm the up-regulation of OXPHOS activity in HO lesions. Notably, an FOP patient case report showed marked fluorodeoxyglucose uptake in PET/CT scan in soft tissue masses without obvious ossification (Kulwin & Binkovitz, 2009), thus supporting our claim that heterotopic cartilage formation requires OXPHOS activation.

We used FOP-iMSCs as a chondrocyte precursor in this in vitro study. Lineage tracing studies identified several precursors of HO in FOP model mice (Dey et al, 2016), one of which is FAPs in muscle tissues (Lees-Shepard et al, 2018). FAPs are involved in the regeneration of injured muscle tissues (Joe et al, 2010), and GSEA in this study also showed the enrichment of genes related to FAPs after pinch injury. It is of interest that this enrichment was downregulated by IACS treatment.

The activation of FAPs may vary across different differentiation processes. Specifically, in chondrogenic differentiation, we propose that FAP activation is characterized by the accumulation of PDGFRα-positive cells and the expression of chondrogenic marker genes. Immunohistochemical studies have demonstrated that FAP activation is linked with enhanced mitochondrial biogenesis. IACS

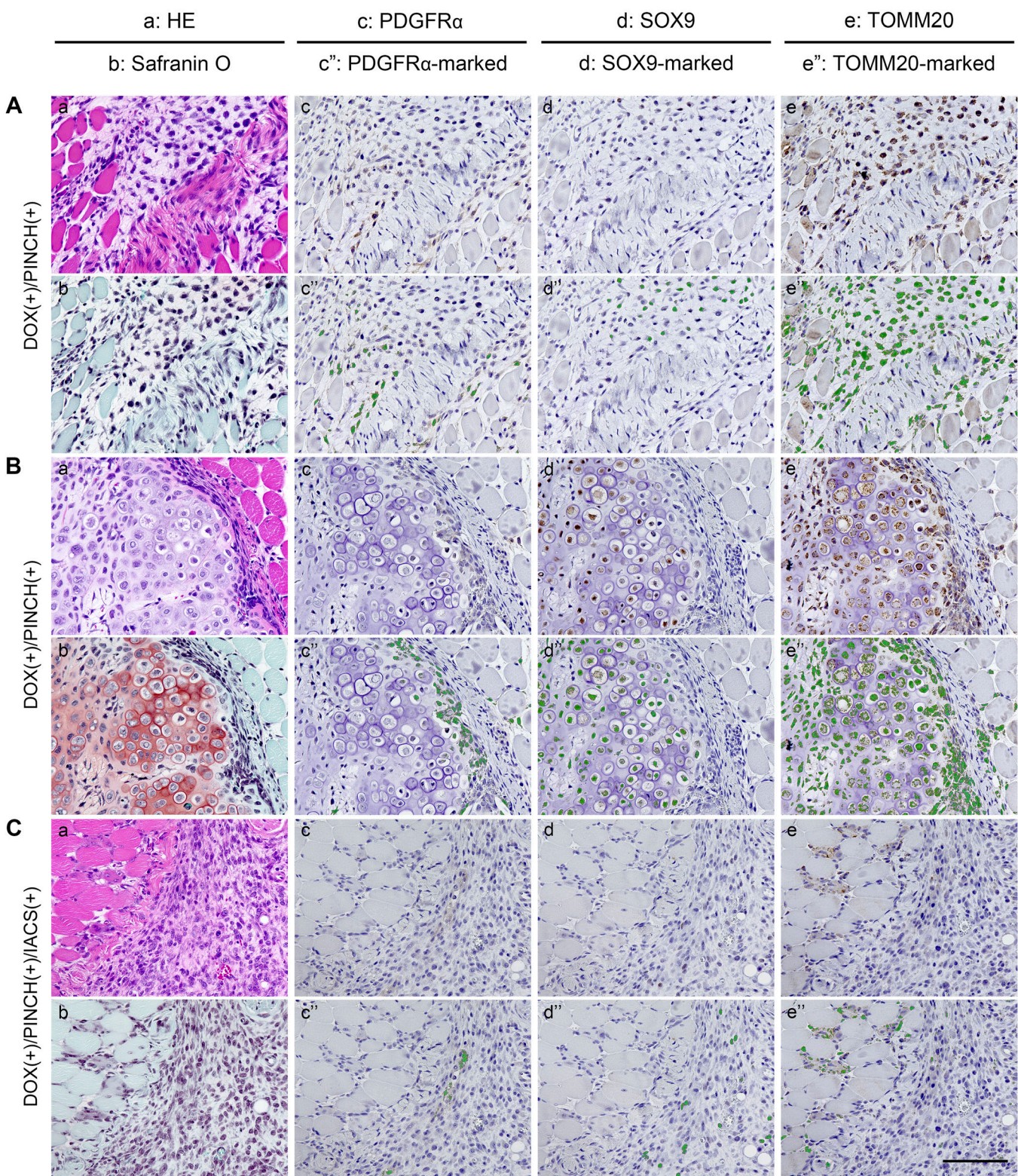

**Day 7 post-injury**

**Figure 7. Inhibition of OXPHOS suppresses the proliferation and differentiation of PDGFRα-positive cells: histological analyses.**
**(A, B, C)** Histological analyses of resected muscle tissues of DOX (+)/PINCH (+) samples (A, B) and DOX (+)/PINCH (+)/IACS (+) samples on the day 7 post-injury (C). Each sample was stained with HE (a), Safranin O and Fast Green (b), antibodies for PDGFRα (c), SOX9 (d), and TOMM20 (e). Cells positive for PDGFRα, SOX9, or TOMM20 were marked and painted in green (c", d", e"). Scale bar = 100 μm.

treatment, while not significantly affecting proliferation or cell death, resulted in a reduced number of PDGFRα-positive cells. Collectively, these findings indicate that FAPs rely on OXPHOS as an energy source for activation. Consequently, IACS treatment inhibits FAP activation, leading to the emergence of cells that are negative for both PDGFRα and SOX9, akin to fibroblastic cells.

We found IACS administration to dramatically reduce *Inhba* expression, which encodes a subunit of Activin A, in pinch-injury lesions (Fig 5E). Activin A, as a cytokine, is secreted by a range of innate immune cells, such as neutrophils, monocytes, macrophages, and dendritic cells, and can be produced by injured soft tissues (Phillips et al, 2009). In the conditional global knock-in mouse model expressing *Acvr1R206H*, Activin A expression was also observed in fibroblastic regions and ectopic chondrocytes (Convente et al, 2018). Although it is unclear which cells in injured tissues are targets of IACS, systemic reduction of this vital cytokine by IACS is beneficial for preventing disease progression, as demonstrated by HO inhibition at uninjured sites in this study. Future experiments using more discriminating and quantitative methods, such as single-cell RNA sequencing, will be vital for further exploration of this mechanism.

Our data indicated that OXPHOS is a critical downstream pathway of mTORC1 in chondrogenesis. However, the mechanism by which OXPHOS induces chondrogenic differentiation is not yet known. OXPHOS is the principal process for ATP production and impacts a wide range of biological processes, including cell growth. OXPHOS contributes to cell growth by promoting aspartate synthesis through its role as an electron acceptor (Birsoy et al, 2015; Sullivan et al, 2015). Cell growth, however, is unnecessary for the induction of chondrogenic differentiation because IACS treatment inhibited differentiation without affecting cell growth. In acquired HO from a burn injury, ATP hydrolysis with apyrase decreased SMAD1/5/8 phosphorylation, thus inhibiting HO formation (Peterson et al, 2014). Considering the importance of BMP signaling in FOP, this mechanism may also contribute to HO inhibition by IACS administration. In 3D pellet formation experiments, IACS treatment significantly inhibited matrix formation. Alpha-ketoglutarate is an essential intermediate of the TCA cycle and a rate-limiting cofactor of prolyl 4-hydroxylase (Johnson et al, 2000). The hydroxylation of proline residues by prolyl 4-hydroxylase serves to stabilize collagen helices (Stegen et al, 2019), whereas OXPHOS inhibition by NO inhibits matrix formation by chondrocytes (Johnson et al, 2000). These data together suggest the involvement of matrix formation as a principal contributing function of OXPHOS in chondrogenic differentiation.

IACS administration markedly reduced HO formation without causing abnormalities in the heart, liver, kidney, and lung (Fig S8A and B). Physiological endochondral ossification, a crucial process for longitudinal growth, occurs in growth plates at the ends of the long tubular bones of growing limbs (Kronenberg, 2003). Disease onset in most FOP patients occurs when the growth plate is still open. Although there is an analogous process in HO, the role of mitochondrial activity differs. Because the growth plate is an avascular tissue, chondrocytes in the growth plates are programmed to function independently from mitochondrial respiration (Yao et al, 2019). HO usually occurs in vascularized and well-oxygenated soft tissues, and develops much faster than the growth plate development, so heterotopic cartilage formation is likely dependent on

OXPHOS. IACS inhibition may thus have little effect on growth plates and may be exploited as an advantage for selective therapeutic application.

Our study highlights the contributions of mitochondrial biogenesis and OXPHOS activity to the initiation and development of HO in FOP, thus expanding the breadth of FOP disease etiology to energy metabolism, which has not received much attention until now. Altogether, we believe targeting OXPHOS is a promising therapeutic strategy for FOP.

# Materials and Methods

## Cell culture

FOP patient-derived iPSCs (FOP-iPSCs) were maintained in primate embryonic stem (ES) cell medium (ReproCELL) supplemented with 4 ng/ml recombinant human FGF2 (Wako Pure Chemical) as described in our previous report (Matsumoto et al, 2013). The induction and maintenance of induced neural crest cells and iMSCs derived from iPSC were previously described (Fukuta et al, 2014). FOP-iPSCs used in this study, FOP-iPSCs from patients 1 and 2, previously described as vFOP4-1 and vFOP5-22 (Matsumoto et al, 2013), harbor an R206H heterozygous mutation in the *ACVR1/ALK2* gene, and gene-corrected resFOP-iPSCs were generated by BAC-based homologous recombination (Matsumoto et al, 2015). All experiments shown in the main figures were performed using FOP-iPSC and resFOP-iPSC of patient 1.

## *FOP-ACVR1* conditional transgenic mice

The establishment of human mutant *ACVR1/ALK2* (FOP-ACVR1) conditional transgenic mice (FOP-ACVR1 mice) was reported previously (Hino et al, 2017), and 18- to 23-wk-old mice were used in the animal experiments. The age and body weight at the initial time point were matched between each group. To conditionally induce the expression of FOP-ACVR1, mice were administered 1 mg/ml DOX and 10 mg/ml sucrose by water drinking. All animal experiments were performed under the principles of the 3Rs and were approved by the Institutional Animal Committee of Kyoto University.

## Chondrogenic induction

Chondrogenic induction was performed following the description in our previous report (Hino et al, 2015; Matsumoto et al, 2015) with some modifications (Ruhl & Beier, 2019). For 2D chondrogenic induction, cells ($6 \times 10^5$/well) were seeded into a fibronectin-precoated 12-well plate, and the chondrogenic induction was initiated after cells were adhered uniformly. For 3D chondrogenic induction, cells ($2.5 \times 10^5$/well) were suspended and transferred to a PrimeSurface 96U plate and centrifuged to form a cell pellet. Chondrogenic induction was carried out using the chondrogenic medium with Activin A (100 ng/ml), with or without IACS-010759 (Selleck), and the medium was changed every 2 d until day 9 or day

21 for 2D and 3D chondrogenic induction, respectively, unless stated otherwise. All cultures were maintained at 37°C under 5% (vol/vol) $CO_2$ and 100% humidity.

### RNA extraction and quantitative PCR

Culture cells were lysed by RLT lysis buffer (QIAGEN) with 1% (vol/vol) 2-mercaptoethanol (Nacalai Tesque), and total RNA was extracted from lysates using RNeasy Mini Kit (QIAGEN) and treated with RNase-Free DNase Set (QIAGEN) to remove genomic DNA. For RNA extraction from muscles, tissues were resected after euthanizing mice by $CO_2$ and homogenized using Multi-Beads Shocker (Yasui Kikai Corporation) according to the manufacturer's instructions. Total RNA was extracted from homogenized tissues using Sepasol-RNA I Super G (Nacalai Tesque) and RNeasy Mini Kit (QIAGEN) and treated with RNase-Free DNase Set (QIAGEN) to remove genomic DNA. RNA concentration was measured using NanoDrop, and 0.5–1 µg of total RNA was reverse-transcribed to single-stranded cDNA using ReverTra Ace RT-qPCR Master Mix (TOYOBO). All RNA and cDNA samples were restored at –80°C for subsequent experiments. Quantitative PCR (RT-qPCR) was performed with SYBR Green Master Mix (TOYOBO) on the StepOnePlus instrument (Applied Biosystems) following the manufacturer's instructions. RT-qPCR data were analyzed according to the $2^{-\Delta\Delta Ct}$ method and are presented as the mean of fold changes compared with the level of the housekeeping gene beta-actin (*ACTB*). The primer sequences used in this study are listed in Table S1.

### mtDNA copy number

Cells were washed with clean PBS buffer and collected in fresh PBS buffer with cell scrapers. Genomic DNA including nuclear DNA and mtDNA was purified using the DNeasy Blood & Tissue Kit (QIAGEN) and treated with RNase A (QIAGEN) to remove RNA and Proteinase K (QIAGEN) to digest proteins for obtaining highly purified DNA samples. A relative mtDNA copy number was analyzed with Human mtDNA Monitoring Primer Set (Takara Bio). Genomic DNA (10 ng) was applied in the PCR system, and data analysis was performed following the manufacturer's manual.

### Alcian Blue staining

Cells were fixed with 4% PFA for 10 min at room temperature. After washing three times with PBS, cells were incubated with 0.1N hydrochloric acid (Muto Pure Chemicals) for 1 min, then stained with Alcian Blue Stain Solution, pH 1.0 (Muto Pure Chemicals) to detect cartilage matrix proteoglycans.

### Glycosaminoglycan (GAG) measurement

GAG content was quantified with Blyscan Glycosaminoglycan Assay Kit (Biocolor), and double-stranded DNA (dsDNA) content was assayed using Quant-iT PicoGreen dsDNA Assay Kit (Thermo Fisher Scientific). All experimental procedures were performed strictly following the manufacturer's manual.

### Extracellular flux analysis

OCR and ECAR were measured using the Seahorse XF96 analyzer (Agilent Technologies). In mitochondrial and glycolysis stress assay during chondrogenic induction, cells (40,000/well) were seeded into a fibronectin-precoated Seahorse 96-well mini plate (Agilent Seahorse XF96). The chondrogenic medium with Activin A (100 ng/ml) was changed every 3 d. Mitochondrial respiration and glycolysis were assayed on day 6 of chondrogenic induction. Cells were allowed to incubate for 30–60 min before the assay in a non-$CO_2$ incubator at 37°C, after which XF Assay Medium supplemented with 10 mM D-glucose, 2 mM L-glutamine, and 1 mM pyruvate (pH 7.4) for mitochondrial stress assay or XF Assay Medium supplemented with 2 mM L-glutamine (pH 7.4) for glycolysis stress assay was changed. All procedures were performed following the manufacturer's manual. The cell culture mini plate was transferred to the Seahorse XF96 analyzer after calibrating the compound-loaded hydrate sensor cartridge. OCR (mitochondrial stress assay) was measured at baseline and followed by sequential injections of oligomycin (2 µM), FCCP (1 µM), and rotenone/antimycin A (0.5 µM), respectively. The basal respiration, maximal respiration, and ATP production were calculated according to the manufacturer's manual. For the glycolysis stress assay, ECAR was measured at baseline and followed by consecutive injections of glucose (10 mM), oligomycin (2 µM), and 2-DG (50 mM), and basal glycolysis and glycolytic ability were calculated. The parallel plate was prepared for dsDNA assay, and the OCR and ECAR data were normalized with the concentration of dsDNA.

Basal respiration is oxygen consumption used to meet cellular ATP demand resulting from mitochondrial proton leak, which shows the energetic demand of the cell under the baseline condition. For more information about ATP production, the decrease in OCR upon injection of the ATP synthase inhibitor oligomycin represents the portion of basal respiration that was being used to drive ATP production, which shows ATP produced by the mitochondria that contributes to meeting the energetic needs of the cell. Maximal respiration is the maximal OCR attained by adding the uncoupler FCCP. FCCP mimics a physiological "energy demand" by stimulating the respiratory chain to operate at maximum capacity. Hence, the maximal respiration shows the maximum rate of respiration that the cell can achieve.

In assays that examine the inhibitors in OCR, chondrogenic differentiated cells (50,000/well) were resuspended by Accutase (Thermo Fisher Scientific)—a cell dissociation reagent—treatment, filtered by 70-µm cell strainers, and reseeded into a fibronectin-precoated Seahorse 96-well mini plate. Cells were treated with each inhibitor for 24 h after they were uniformly adhered, and then, OCR and ECAR were measured as described above.

### Pinch-injury operation in gastrocnemius and sample collection

FOP-ACVR1 mice were anesthetized with isoflurane (5% for induction, 2–3% for maintenance) (Pfizer). The skin was incised at the lateral surface of the hindlimb, to expose the gastrocnemius. The middle portion of the gastrocnemius was pinched by tissue forceps, and pressure was applied for 5 s, and then, the skin was closed and sutured. A sham operation was performed by skin incision and

suture only without gastrocnemius pinch injury. On days 7 and 21 after the pinch-injury operation, mice were euthanized by $CO_2$, and the heart, lung, liver, kidney, and gastrocnemius muscle at the injured sites were harvested and fixed with 4% PFA or stored at −80°C.

### IACS-010759 administration

IACS-010759, dissolved in 10% DMSO with 0.5% (wt/vol) methyl-cellulose 400, was administered orally once a day from 2 d before pinch injury. Mice in the vehicle-treated group were administered the same solution without IACS-010759.

### Micro-computed tomography (μCT) scan

Mice were anesthetized with a mixture of medetomidine, mid-azolam, and butorphanol, and scanned with an X-ray μCT system (inspeXio SMX-100CT; Shimadzu), and three-dimensional images were obtained by TRI/3D-BON software (Ratoc System Engineering Co., Ltd.) according to the manufacturer's instructions. The scan was performed by an experimenter blinded to the experimental groups.

### Immunoblotting

Cell samples were lysed in an SDS buffer with protease and phosphatase inhibitors (Pierce) by vortexing for 30 min. Protein extracts were diluted twofold in Laemmli buffer and heated for 10 min at 95°C before SDS–PAGE and subsequent electrophoretic transfer to a PVDF membrane. After 1 h of blocking in 5% non-fat dry milk in TBS, the PVDF membrane was incubated in primary antibody overnight at 4°C. The PVDF membrane was then washed thoroughly with TBS containing 0.1% Tween-20 (TBST) three times, incubated for 1 h at room temperature in a secondary antibody, and thoroughly washed three times again with TBST before imaging on ChemiDoc Touch MP Imaging System (BIO-RAD). Primary antibodies and secondary antibodies are listed in Table S2.

### Histology

Tissue samples were collected in a sterile environment, fixed with 4% PFA immediately, embedded in paraffin, and sectioned (5-μm sections). Sections were deparaffinized, rehydrated, and stained with hematoxylin and eosin (HE), Safranin O, and von Kossa.

### Immunohistochemistry

Tissue samples embedded in paraffin were sectioned by sequential slicing. For antigen retrievals, the 5-μm tissue sections were incubated in Tris–EDTA (pH 9.0; Abcam) for 20 min at 90°C–95°C after dewaxing. After washing, the sections were incubated with 3% $H_2O_2$ for 10 min at room temperature. To avoid non-specific binding, the sections were blocked with Animal-Free Blocking Solution (15019; CST) for 1 h at room temperature. The primary antibodies were performed overnight at 4°C in a humidifier box. Antibody binding was revealed using species-specific secondary antibodies conjugated with HRP for 1 h, followed by DAB staining (8059; CST) for 5–8 min at room temperature. Sections were counterstained with

hematoxylin (S3309; Dako). After dehydration, sections were sealed in SignalStain Mounting Medium (14177; CST). Finally, the sections were observed under a fluorescence microscope (BZ-X800; Key-ence). For negative controls, the primary antibody was omitted. Negative controls did not show staining. Concentrations and sources of primary and secondary antibodies are described in Table S2.

### RNA-seq for transcriptome profiling

Total RNA was prepared from tissue samples as described. RNA integrity was assessed with 4200 TapeStation (Agilent Technologies). The library preparation was performed using TruSeq Stranded mRNA Sample Prep Kit (Illumina) according to the manufacturer's instructions. The libraries were sequenced with Illumina NextSeq 550. Adapter sequences and low-quality bases were trimmed from the raw reads using Cutadapt v3.4 (Martin, 2011). The trimmed reads were mapped to human (hg38) and mouse (mm10) reference genome sequences using STAR ver 2.7.9a (Dobin et al, 2013) with the GENCODE (v36 for human and v23 for mouse) gtf file (Frankish et al, 2019). The number of reads mapped to each gene (raw read counts) was calculated using htseq-count ver. 0.13.5 with the GENCODE gtf file (Anders et al, 2015). Gene expression levels were determined as Transcripts Per Kilobase Million (TPM) values with DESeq2 v1.30.1 (Love et al, 2014). GSEA was performed with GSEA_4.1.0.

### Gas chromatography–mass spectrometry (GC-MS) analysis of metabolites

Cultured cells were washed twice with 4 ml of ice-cold PBS (−). Metabolism of cells was quenched with 1 ml of prechilled extraction solution and chilled at −30°C for 5 min. The extraction solution was made by mixing 80% of aqueous methanol and the internal standard solution (2-isopropylmalic acid, 0.1 mg/ml) at a ratio of 250:6. Cells were collected using a cell scraper and transferred to a sample tube. After centrifugation at 9,000$g$ for 5 min at 4°C, the supernatant was collected. The cell pellet was resuspended in 0.5 ml of extraction solution and centrifuged at 9,000$g$ for 5 min at 4°C. The supernatants were pooled and dried in a centrifugal vacuum evaporator (CVE-3100; EYELA) and stored at −80°C until sample analysis.

The dried sample was dissolved in 80 μl of methoxyamine solution (20 mg/ml in pyridine) and agitated at 1,200 rpm for 30 min at 37°C (Eppendorf ThermoMixer C). After the addition of 40 μl of MSTFA (N-methyl-N-TMS-trifluoroacetamide), the samples were agitated at 1,200 rpm for 30 min at 30°C. The samples were centrifuged at 3,000$g$ for 5 min, and the supernatants were subjected to GC-MS analysis.

GC-MS analysis was performed with GC-MS-QP2010 Ultra (Shimadzu Corporation) (Jin et al, 2015). After an initial time of 2 min at 80°C, the temperature was increased to 147°C at a rate of 15°C/min, followed by an additional constant temperature period at 330°C for 5 min. The mass spectrometer conditions were as follows: electron ionization mode with an ionization voltage of 70 eV, ion source temperature of 200°C, and interface temperature of 250°C. The derivatized metabolites were separated on a DB-5 column (30 m × 0.25 mm I.D.). Full-scan mass spectra were acquired from m/z 85 to

460. Data acquisition and peak processing were performed using GC-MS solution software, version 2.71 (Shimadzu). All the data of metabolites were normalized with the concentration of dsDNA of each sample.

## Statistics

Statistical analysis was performed using GraphPad Prism 9 (GraphPad Software). All numerical values are presented as mean values ± SEM. The statistical significance of all experiments was calculated using a two-tailed $t$ test, two-way ANOVA for Tukey's multiple comparisons test or Šidák's multiple comparisons test, or one-way ANOVA for Tukey's multiple comparisons test. $P$-values of less than 0.05 were considered statistically significant. No randomization was used to allocate animals to groups, and the investigators were not blinded to experimental groups during the analysis.

## Data Availability

Raw data from this study were submitted under the NCBI GEO Series accession number GSE220725 for mouse muscle samples and GSE221128 for human cells.

## Supplementary Information

## Acknowledgements

We thank Kazusa Okita and Satoko Sakurai for their technical advice in RNA sequencing, Masamichi Nakatake for his technical support, and Kelvin Hui for revising and proofreading the article. We also thank the JSPS research fellowship for young scientists (DC1) in Japan awarded to L Sun. This work was supported by a Grant-in-Aid for the Acceleration Program for Intractable Disease Research Utilizing Disease-Specific iPS Cells from Japan Agency for Medical Research and Development (AMED) (22bm0804006) to J Toguchida, the Core Center for iPS Cell Research (22bm0104001) to S Kawai, Y Jin, and J Toguchida, the Centers for Clinical Application Research on Specific Disease/ Organ (type B) grants (22bm0304004) to J Toguchida, Grants-in-Aid for Scientific Research from Japan Society for the Promotion of Science (JSPS) (18H02928) to Y Jin and J Toguchida, and (22K19600) to Y Jin, the iPS Cell Research Fund to J Toguchida, and Grant-in-Aid for JSPS Fellows from JSPS (20J23707) and the iPS Cell Research Fund to L Sun. These funders had no role in the study design, data collection and analysis, decision to publish, or preparation of the article.

## Author Contributions

L Sun: data curation, formal analysis, validation, investigation, visualization, methodology, and writing—review and editing.
Y Jin: conceptualization, data curation, formal analysis, supervision, funding acquisition, validation, investigation, visualization, methodology, project administration, and writing—original draft, review, and editing.
M Nishio: data collection.
M Watanabe: GC-MS analysis.
T Kamakura: data collection.
S Nagata: data collection.
M Fukuda: data collection.
H Maekawa: guidance on the mouse model.
S Kawai: validation and writing—review and editing.
T Yamamoto: technical support for bulk RNA sequencing.
J Toguchida: conceptualization, resources, supervision, funding acquisition, visualization, project administration, and writing—original draft, review, and editing.

## Conflict of Interest Statement

The authors declare that they have no conflict of interest.

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
