## [Reviewer comments · Life Science Alliance]

Life Science Alliance

Oxidative phosphorylation is a pivotal therapeutic target of fibrodysplasia ossificans progressiva

Liping Sun, Yonghui Jin, Megumi Nishio, Makoto Watanabe, Takeshi Kamakura, Sanae Nagata, Masayuki Fukuda, Hirotsugu Maekawa, Shunsuke Kawai, Takuya Yamamoto, and Junya Toguchida

DOI: <https://doi.org/10.26508/lsa.202302219>

Corresponding author(s): Yonghui Jin, Institute for Life and Medical Sciences, Kyoto University and Junya Toguchida, Center for iPS Cell Research and Application, Kyoto University

Review Timeline:

Submission Date:	2023-06-15
Editorial Decision:	2023-08-24
Revision Received:	2024-01-10
Editorial Decision:	2024-01-29
Revision Received:	2024-01-31
Accepted:	2024-02-07

Transaction Report:

August 24, 2023

Re: Life Science Alliance manuscript #LSA-2023-02219-T

Dr. Yonghui Jin
Institute for Life and Medical Sciences, Kyoto University
JAPAN

Dear Dr. Jin,

Thank you for submitting your manuscript entitled "Oxidative phosphorylation is a pivotal therapeutic target of fibrodysplasia ossificans progressiva" to Life Science Alliance. The manuscript was assessed by expert reviewers, whose comments are appended to this letter. We invite you to submit a revised manuscript addressing the Reviewer comments.

Thank you for this interesting contribution to Life Science Alliance. We are looking forward to receiving your revised manuscript.

Sincerely,

B. MANUSCRIPT ORGANIZATION AND FORMATTING:

Reviewer #1 (Comments to the Authors (Required)):

This interesting manuscript by Lipin Sun et al seeks to establish that OXPPOS is essential for chondrogenesis and HO formation in FOP. Using FOP patient derived iPSC they demonstrate that Activin A drives chondrogenesis in the FOP-iMSC cells and this is associated with increased mitochondrial biogenesis and activity. Inhibition of mTOR using Rapamycin prevented the changes in mitochondrial activity induced by Activin A. Importantly, a complex 1 inhibitor prevented chondrocyte differentiation in vitro and could reduce HO induction in their PINCH model. Strengths of this manuscript include the rigorous experimental design, convincing data showing increased mitochondrial biogenesis and OXPPOS during chondrogenesis and the convincing inhibition of HO formation in their PINCH model. The mechanistic connection between mTORC1 and OXPPOS is tenuous at best. I have the following specific comments/questions.

1. The data linking mTOR to OXPPOS is weak and circumstantial. mTOR inhibition has broad effects beyond the direct inhibition of OXPPOS. Rapamycin directly inhibits protein synthesis in many systems and likely inhibits chondrogenic differentiation directly. Differentiation and protein and matrix production are energetically demanding processes. Inhibiting either should reduce energetic demand and reduce OXPPOS. Does inhibiting differentiation or protein synthesis independent of changes in mTOR activation limit OXPPOS? They should include a more thorough investigation of the link between mTOR and OXPPOS or soften this conclusion.
2. Inhibiting complex 1 using the IACS drug inhibits chondrocyte differentiation in vitro and in vivo. Does IACS affect mTORC1 activity? The authors should verify the specificity IACS on chondrocyte differentiation using an independent ETC inhibitor.
3. In Fig 3, the pellet cultures are much smaller in IACS treated pellets. The authors attribute this to inhibitor effects on matrix production, not growth. They should evaluate proliferation and apoptosis. Is cell size affected? Are these cells able to go through hypertrophy?
4. A major function of mitochondrial activity is to support nucleotide biosynthesis for proliferation. The authors claim inhibiting OXPPOS suppresses proliferation and differentiation of PDGFRa positive cells. However, they provide no data looking at proliferation directly. They should evaluate proliferation using either BRDU or phosphor-Histone H3 in the PDGFRa+ MSC in the PINCH model. Likewise, they should evaluate apoptosis in these cells using TuNEL or some other appropriate method.
5. Mitochondrial function is dispensable for chondrocyte differentiation as chondrocyte specific TFAM knockout mice have almost normal endochondral ossification (PMID:31105007). Why do the authors think the FOP cells are unique in that they require OXPPOS when normal chondrocytes don't? This should be added to the discussion.

Reviewer #2 (Comments to the Authors (Required)):

The manuscript by Sun et al describes the critical role of mTORC1 signaling for mitochondrial oxidative phosphorylation in activin A-induced chondrogenesis in cellular and animal models of FOP, connecting cellular energy metabolism to endochondral ossification. The authors provided the evidence of metabolism data and transcriptome analysis data. Overall, the manuscript was clearly written and data presentation in the figures are easy to follow. But the current study can benefit from additional data on how mTORC1 activation for OXPPOS activity and mitochondrial biogenesis.

For instance, 1) How is the current study associated with the authors' previous finding that ENPP2 is a molecular linker for FOP-ACVR1 and mTORC1 signaling in FOP?

2) Would Enpp2 inhibitor or Enpp2 knockdown replicate the results with rapamycin or IACS-010759?

Reviewer #3 (Comments to the Authors (Required)):

The goal of this study is to examine how activated mTORC1 induces precursor cells for chondrogenic differentiation. RNAseq analyses identified enriched expression of genes relevant to oxidative phosphorylation in response to activin A-induced chondrogenesis of Acvr1-R206H mesenchymal cells (derived from human patient iPSCs). Functional assays indicate increased oxidative phosphorylation and increased mitochondrial biogenesis. Further, the mTORC1 inhibitor rapamycin reduced oxidative phosphorylation in Acvr1-R206H cells. The oxidative phosphorylation inhibitor IACS-010759 inhibited in vivo cartilage matrix production, the expression of FAP-associated gene expression, and HO in transgenic Acvr1-R206H mice. These data support that oxidative phosphorylation is a downstream mediator of mTORC1 signaling during chondrogenesis in cells with the FOP Acvr1-R206H mutation.

Comments to authors

This manuscript continues the authors' investigation of mTORC1 signaling in heterotopic ossification and FOP, and reports important but little investigated effects on cell metabolism and its impact on HO formation.

Specific comments:

1. Page 9-10/Figure 2. The authors are requested to provide additional explanation for OCR analyses used (basal mitochondrial respiration, maximal mitochondrial respiration, ATP production) in Results and/or Methods.
Figure 2 data: FOP cells seem to have higher mitochondrial respiration at day 0, prior to induction with activin A. The authors are asked to include comments about this.
Have the authors considered or investigated whether the cell metabolic response is dependent on or specific to activin A, or is the response associated with chondrogenesis through other induction factors as well?
2. Chondrogenesis would seem to be a metabolically demanding process - are the authors surprised that the control cells didn't also show increased oxphos activity during chondrogenesis?
Wouldn't ICS (Figure 3B,C,D) be expected to also inhibit control cell chondrogenesis? Or are the authors suggesting that the chondrogenic differentiation is completely different in the control and mutant cells? Please clarify.
3. Page 12/Figure 4. Centralized nuclei can be detected in both damaged and regenerating muscle fibers - this is not a reliable indicator of 'vigorous' regeneration in the absence of other markers.
4. Page 15/Figure 6. The authors are requested to include more information about the FAP data set used as a standard for comparison to their data - including source of data and the state of FAPs (injured or non-injured? activated or quiescent FAPs?)
Note that if the 'standard' data set is derived from analysis of whole muscle, this seems to be a somewhat difficult comparison to make since FAPs are a very small population of cells within the muscle tissue.
5. The Abstract/Intro suggest that FAPs were directly examined in this study - this need to be clarified to avoid mis-leading the reader.
6. Figure 6E, F, G - The timepoint/day post injury that is shown/analyzed should be clearly indicated.
Would be helpful to provide additional panels labeling to indicated the antibodies used in this figure.
7. The authors are asked for further explanation:
IACS inhibits chondrogenesis, but fibroblasts persist; however, genes associated with FAPs (with activated FAPS?) persist.
Are the noted FAP-associated genes expressed in response to FAP activation? This was not clear in the text.

Dear Editor,

We have revised the manuscript according to the reviewers' comments. In particular:

-We have added a new Figure S2 to answer reviewer #1 comment 1. The effects of protein synthesis inhibitors on OCR were shown.

-We have added a new Figure S4 to answer reviewer #1 comment 2. The effect of metformin on OCR and chondrogenesis, another inhibitor of mitochondrial respiratory complex I, was shown here. The effects of IACS and metformin on mTORC1 activity were also shown in this figure.

-We have added a new Figure S9 to answer reviewer #1 comment 4. We performed immunohistochemistry with Ki67 and pHH3 antibodies –makers of proliferation, and cleaved caspase 3 antibody-a marker of apoptosis.

-Aiming to answer reviewer #2 comment, we performed OCR analysis after ENPP2 inhibitor treatment. This result was presented in this letter.

-We have added a new Figure S5 to answer reviewer #3 comment 3. We confirmed the skeletal muscle regeneration by immunohistochemistry with myozenin 1 antibody.

All the changes made to the text are highlighted in red.

Following is the point-by-point answers to the reviewers' comments:

Reviewer #1

The mechanistic connection between mTORC1 and OXPHOS is tenuous at best. I have the following specific comments/questions.

1. The data linking mTOR to OXPHOS is weak and circumstantial. MTOR inhibition has broad effects beyond the direct inhibition of OXPHOS. Rapamycin directly inhibits protein synthesis in many systems and likely inhibits chondrogenic differentiation directly. Differentiation and protein and matrix production are energetically demanding processes. Inhibiting either should reduce energetic demand and reduce OXPHOS. Does inhibiting differentiation or protein synthesis independent of changes in mTOR

activation limit OXPHOS? They should include a more thorough investigation of the link between mTOR and OXPHOS or soften this conclusion.

To verify the impact of inhibiting protein synthesis, we initiated chondrogenesis in induced mesenchymal stem cells (iMSCs) over a period of four days, subsequently treating them with cycloheximide or puromycin for 24 hours before conducting an OCR assessment (refer to new Figure S2). Notably, the suppression of protein synthesis markedly reduced OCR parameters. While the precise molecular interplay between mTORC1 and OXPHOS remains elusive, our findings suggest that protein synthesis is one of the downstream effectors of mTORC1, playing a role in the regulation of OXPHOS. We have accordingly moderated the conclusions drawn from these results.

2. Inhibiting complex 1 using the IACS drug inhibits chondrocyte differentiation in vitro and in vivo. Does IACS affect mTORC1 activity? The authors should verify the specificity IACS on chondrocyte differentiation using an independent ETC inhibitor.

We have conducted the suggested experiments using another inhibitor of mitochondrial complex I, metformin. Treatment with metformin led to a reduction in OCR within 24 hours, which was followed by an inhibition of chondrogenesis in FOP-iMSCs in vitro (refer to new Figure S4, A and B). Furthermore, both IACS and metformin exhibited no impact on the phosphorylation of S6 and p70S6K (refer to new Figure S4C). This observation indicates that inhibiting complex I does not influence mTORC1 activity. These findings underscore the critical role of OXPHOS in the chondrogenesis of FOP-iMSCs.

3. In Fig 3, the pellet cultures are much smaller in IACS treated pellets. The authors attribute this to inhibitor effects on matrix production, not growth. They should evaluate proliferation and apoptosis.

In the chondrogenic pellet culture, treatment with IACS resulted in a decrease in Safranin O staining and glycosaminoglycan (GAG) production. However, the DNA

content, which serves as an indicator of cell quantity, remained unchanged. This suggests that IACS did not significantly impact cell proliferation. This notion is further corroborated by in vivo data presented in response to comment 4, as depicted in new Figure S9.

Is cell size affected? Are these cells able to go through hypertrophy?

In our system, achieving complete hypertrophic differentiation of chondrocytes necessitated an extended culture period with thyroid hormone (T3) treatment (PMID: 33636111). However, a small number of cells in the vehicle group exhibited hypertrophic morphology, a characteristic not observed in the IACS-treated group. This observation suggests that IACS may influence cell size by inhibiting hypertrophic differentiation.

4. A major function of mitochondrial activity is to support nucleotide biosynthesis for proliferation. The authors claim inhibiting OXPHOS suppresses proliferation and differentiation of PDGFR α positive cells. However, they provide no data looking at proliferation directly. They should evaluate proliferation using either BRDU or phosphor-Histone H3 in the PDGFR α MSC in the PINCH model. Likewise, they should evaluate apoptosis in these cells using TuNEL or some other appropriate method.

Responding to the reviewer's suggestion, we conducted immunohistochemical staining using Ki67 and phospho-histone H3 (pHH3) to assess cell proliferation. Additionally, apoptosis was evaluated with the aid of a cleaved caspase 3 antibody (refer to new Figure S9). No significant differences in proliferation or apoptosis were detected between the DOX(+)/PINCH(+) and DOX(+)/PINCH(+)/IACS(+) groups. This indicates that IACS primarily inhibits the chondrogenic differentiation of PDGFR α -positive cells, rather than affecting cell death or proliferation. Consequently, we have removed the term "proliferation" from the heading "Inhibition of OXPHOS suppresses the proliferation and differentiation of PDGFR α -positive cells" to reflect these findings more accurately.

5. Mitochondrial function is dispensable for chondrocyte differentiation as chondrocyte specific TFAM knockout mice have almost normal endochondral ossification (PMID:31105007). Why do the authors think the FOP cells are unique in that they require OXPHOS when normal chondrocytes don't? This should be added to the discussion.

While both processes exhibit endochondral ossification, there are notable differences in the environments of growth plate formation during development and HO in soft tissues. HO in patients with FOP typically occurs in vascularized and well-oxygenated soft tissues, contrasting with the avascular nature of the growth plate. Additionally, HO develops at a significantly faster rate compared to growth plate development, potentially necessitating a greater energy expenditure. We have revised the relevant description in the Discussion section on Page 23 to clarify this distinction more effectively.

Reviewer #2

But the current study can benefit from additional data on how mTORC1 activation for OXPHOS activity and mitochondrial biogenesis.

For instance, 1) How is the current study associated with the authors' previous finding that ENPP2 is a molecular linker for FOP-ACVR1 and mTORC1 signaling in FOP? 2) Would Enpp2 inhibitor or Enpp2 knockdown replicate the results with rapamycin or IACS-010759?

Thank you for referencing our prior research. To explore the relationship between ENPP2 and OXPHOS, we induced chondrogenic differentiation in both FOP-iMSCs and resFOP-iMSCs for four days, followed by a 24-hour treatment with the ENPP2 inhibitor, HA130, which was used in our previous study (PMID: 28758906). The treatment with HA130 significantly reduced the OCR in FOP-iMSCs, but not in resFOP-iMSCs (see figures below). This suggests that inhibiting ENPP2 mimics the effects of rapamycin or IACS treatment on OXPHOS. These findings reinforce our previous conclusion that ENPP2 acts as a molecular bridge linking FOP-ACVR1 to mTORC1 activity.

[Figure removed by editorial staff per authors' request]

Reviewer #3

Comments to authors

This manuscript continues the authors' investigation of mTORC1 signaling in heterotopic ossification and FOP, and reports important but little investigated effects on cell metabolism and its impact on HO formation.

Specific comments:

1. Page 9-10/Figure 2. The authors are requested to provide additional explanation for OCR analyses used (basal mitochondrial respiration, maximal mitochondrial respiration, ATP production) in Results and/or Methods.

Following the reviewer's suggestion, we have supplemented our manuscript with additional explanation regarding the Oxygen Consumption Rate (OCR) analyses. These elaborations have been incorporated both in the Results and Methods sections for enhanced clarity and thoroughness.

Figure 2 data: FOP cells seem to have higher mitochondrial respiration at day 0, prior to induction with activin A. The authors are asked to include comments about this.

We currently lack a definitive explanation for these results. In the case of FOP-iMSCs, Activin A activates the TGF β pathway via ACVR1B/ALK4 and also stimulates the BMP pathway through mutant ACVR1A/ALK2. Consequently, Activin A alone is capable of upregulating OXPPOS activity in these cells. However, in resFOP-iMSCs, Activin A does not trigger the BMP signal, thereby failing to enhance OXPPOS activity. We hypothesize that the fetal bovine serum used in the culture medium may contain Activin A (PMID: 19308291), which potentially activates signals through the mutant ACVR1 in FOP cells. This could lead to the activation of OXPPOS even without the addition of exogenous Activin A.

Have the authors considered or investigated whether the cell metabolic response is dependent on or specific to activin A, or is the response associated with chondrogenesis through other induction factors as well.

As of day 0, chondrogenesis had not commenced in FOP-iMSCs, as indicated by the expression levels of marker genes such as SOX9 (refer to Figure 1A). Consequently, the observed upregulation of OXPPOS is independent of chondrogenesis triggered by factors other than Activin A. We have not conducted investigations into the impact of other factors on the OXPPOS activity in FOP-iMSCs. However, it is plausible that if there are factors besides Activin A capable of simultaneously inducing both TGF β and BMP signals in FOP-iMSCs, similar metabolic responses would be observed.

2. Chondrogenesis would seem to be a metabolically demanding process - are the authors surprised that the control cells didn't also show increased oxphos activity during chondrogenesis? Wouldn't ICS (Figure 3B,C,D) be expected to also inhibit control cell chondrogenesis? Or are the authors suggesting that the chondrogenic differentiation is completely different in the control and mutant cells? Please clarify.

In our monolayer chondrogenic differentiation system (as illustrated in Figure 3, B, C, and D), mutant cells (FOP cells) demonstrated robust chondrogenic differentiation in response to Activin A alone. In contrast, the control cells (resFOP cells) showed no

such differentiation. This disparity makes it understandable why the control cells did not exhibit increased OXPHOS activity. Given that Activin A activates both TGF β and BMP signaling pathways in mutant cells, but only the TGF β pathway in control cells, we propose that the chondrogenic differentiation of control cells might necessitate additional BMP ligands in addition to Activin A. We suggest that both control and mutant cells follow a similar chondrogenic differentiation pathway, which requires the activation of both TGF β and BMP signaling.

3. Page 12/Figure 4. Centralized nuclei can be detected in both damaged and regenerating muscle fibers - this is not a reliable indicator of 'vigorous' regeneration in the absence of other markers.

We are grateful for the reviewer's suggestion. To assess muscle fiber regeneration, we conducted an immunohistochemical analysis of myozenin 1 (MYOZ1) (refer to new Figure S5). MYOZ1 is recognized as a protein associated with regeneration (PMID: 32391357). Our findings revealed that myozenin 1 was not expressed in some centrally nucleated myofibers. This observation supports the reviewer's insight that the central localization of nuclei is not a definitive marker of regeneration. Accordingly, we have updated the description in the results section on Page 13 to reflect this new understanding.

4. Page 15/Figure 6. The authors are requested to include more information about the FAP data set used as a standard for comparison to their data - including source of data and the state of FAPs (injured or non-injured? activated or quiescent FAPs?) Note that if the 'standard' data set is derived from analysis of whole muscle, this seems to be a somewhat difficult comparison to make since FAPs are a very small population of cells within the muscle tissue.

The gene set representing FAPs utilized in our study was determined via single-cell RNA sequencing analysis of mononuclear cells isolated from non-injured skeletal muscle (PMID: 31937892). Given that this gene set was identified at single-cell resolution, we believe that the gene signature observed in bulk RNA sequencing is

indicative of the cell number of FAPs. We have included further information regarding this in the results section on Page 17.

5. The Abstract/Intro suggest that FAPs were directly examined in this study - this need to be clarified to avoid mis-leading the reader.

We apologize for any confusion caused earlier. To address this, we have revised the relevant content in both the Abstract and the Introduction sections for greater clarity and accuracy.

6. Figure 6E, F, G - The timepoint/day post injury that is shown/analyzed should be clearly indicated. Would be helpful to provide additional panels labeling to indicated the antibodies used in this figure.

In line with the reviewer's suggestion, we have included details regarding the timepoint and the specific antibodies used in Figure 6, E, F, and G.

7. The authors are asked for further explanation:

IACS inhibits chondrogenesis, but fibroblasts persist; however, genes associated with FAPs (with activated FAPS?) persist.

Are the noted FAP-associated genes expressed in response to FAP activation?
This was not clear in the text.

The genes associated with FAPs are typically expressed in these cells even in their quiescent state. As we addressed in response to comment 4, we believe that the intensity of the FAP gene set signatures is indicative of the number of FAP cells. Therefore, the upregulation of this gene signature following PINCH injury is attributed to the proliferation of FAPs. Treatment with DOX further amplified the number of cells expressing the FAP signature. Conversely, IACS treatment counteracted the effects of DOX, leading to a reversion of the FAPs signature expression level to that observed in DOX(-)/PINCH(+) samples. These points have been elucidated in both the Results (Page 17) and Discussion (Page 21) sections of our paper.

January 29, 2024

RE: Life Science Alliance Manuscript #LSA-2023-02219-TR

Dr. Yonghui Jin
Institute for Life and Medical Sciences, Kyoto University
Department of Regeneration Sciences and Engineering
53 Kawahara-cho, Shogoin, Sakyo-ku, Kyoto, 606-8507
Kyoto 606-8507
Japan

Dear Dr. Jin,

Thank you for submitting your revised manuscript entitled "Oxidative phosphorylation is a pivotal therapeutic target of fibrodysplasia ossificans progressiva". We would be happy to publish your paper in Life Science Alliance pending final revisions necessary to meet our formatting guidelines.

- please be sure that the authorship listing and order is correct
- please upload your main figures as single files, i.e., one file per figure
- there is no need to upload the Summary blurb as a separate file -- please remove
- please add ORCID ID for the secondary corresponding author -- they should have received instructions on how to do so
- please add the Twitter handle of your host institute/organization as well as your own or/and one of the authors in our system
- we encourage you to revise the figure legend for figure S3 such that the figure panels are introduced in an alphabetical order
- please add callouts for Figures S6A-B and S9A-D to your main manuscript text;
- please move the RNA-seq accession information to a new Data Availability statement at the end of the Materials and Methods section
- the contributions selected for Shunsuke Kawai do not qualify a contributor for authorship. Please either update the contributions in our system and in the Author Contributions section of the manuscript, or let us know if the author needs to be removed.

A. FINAL FILES:

B. MANUSCRIPT ORGANIZATION AND FORMATTING:

Sincerely,

Reviewer #1 (Comments to the Authors (Required)):

In this revised manuscript, the authors have done an outstanding job addressing my comments. I am convinced of their findings. I especially appreciate the thoughtful explanation on the different requirements for OXPHOS during embryonic development and HO formation.

Reviewer #2 (Comments to the Authors (Required)):

The manuscript by Sun et al describes the critical role of mTORC1 signaling for mitochondrial oxidative phosphorylation in activin A-induced chondrogenesis in cellular and animal models of FOP, connecting cellular energy metabolism to endochondral ossification. The authors provided the evidence of metabolism data and transcriptome analysis data. Overall, the manuscript was clearly written and data presentation in the figures are easy to follow. But the current study can benefit from additional data on how mTORC1 activation for OXPHOS activity and mitochondrial biogenesis. For instance,

1) How is the current study associated with the authors' previous finding that ENPP2 is a molecular linker for FOP-ACVR1 and mTORC1 signaling in FOP?

2) Would Enpp2 inhibitor or Enpp2 knockdown replicate the results with rapamycin or IACS-010759?

February 7, 2024

RE: Life Science Alliance Manuscript #LSA-2023-02219-TRR

Dr. Yonghui Jin
Institute for Life and Medical Sciences, Kyoto University
Department of Regeneration Sciences and Engineering
53 Kawahara-cho, Shogoin, Sakyo-ku, Kyoto, 606-8507
Kyoto 606-8507
Japan

Dear Dr. Jin,

Thank you for submitting your Research Article entitled "Oxidative phosphorylation is a pivotal therapeutic target of fibrodysplasia ossificans progressiva". It is a pleasure to let you know that your manuscript is now accepted for publication in Life Science Alliance. Congratulations on this interesting work.

DISTRIBUTION OF MATERIALS:

Again, congratulations on a very nice paper. I hope you found the review process to be constructive and are pleased with how the manuscript was handled editorially. We look forward to future exciting submissions from your lab.

Sincerely,
